# DNA mechanotechnology reveals that integrin receptors apply pN forces in podosomes on fluid substrates

Roxanne Glazier [1], Joshua M. Brockman[1], Emily Bartle[2], Alexa L. Mattheyses [2], Olivier Destaing[3] & Khalid Salaita [1,4]

Podosomes are ubiquitous cellular structures important to diverse processes including cell invasion, migration, bone resorption, and immune surveillance. Structurally, podosomes consist of a protrusive actin core surrounded by adhesion proteins. Although podosome protrusion forces have been quantified, the magnitude, spatial distribution, and orientation of the opposing tensile forces remain poorly characterized. Here we use DNA nanotechnology to create probes that measure and manipulate podosome tensile forces with molecular piconewton (pN) resolution. Specifically, Molecular Tension-Fluorescence Lifetime Imaging Microscopy (MT-FLIM) produces maps of the cellular adhesive landscape, revealing ring-like tensile forces surrounding podosome cores. Photocleavable adhesion ligands, breakable DNA force probes, and pharmacological inhibition demonstrate local mechanical coupling between integrin tension and actin protrusion. Thus, podosomes use pN integrin forces to sense and respond to substrate mechanics. This work deepens our understanding of podosome mechanotransduction and contributes tools that are widely applicable for studying receptor mechanics at dynamic interfaces.

[1] Wallace H. Coulter Department of Biomedical Engineering, Georgia Institute of Technology and Emory University, Atlanta, GA, USA. [2] Department of Cell, Developmental, and Integrative Biology, University of Alabama at Birmingham, Birmingham, AL, USA. [3] Institute for Advanced Biosciences, Centre de Recherche Université Grenoble Alpes, Inserm U 1209, CNRS UMR 5309, Grenoble, France. [4] Department of Chemistry, Emory University, Atlanta, GA, USA. Correspondence and requests for materials should be addressed to O.D. (email: olivier.destaing@univ-grenoble-alpes.fr) or to K.S. (email: k.salaita@emory.edu)

Podosomes are specialized acto-adhesive structures that coordinate a variety of cell-type specific processes ranging from forming the sealing zone for bone resorption in osteoclasts to facilitating migration and antigen scavenging in immune cells[1–4]. In Wiskott–Aldrich syndrome, cells fail to form podosomes, and patients experience frequent infections, impaired blood clotting, and altered bone resorption[5–8]. In HIV, however, numerous enlarged podosomes are associated with increased cell migration, macrophage tissue infiltration, and elevated bone degradation[9,10]. Invadopodia, which are structurally similar to podosomes, facilitate cancer cell migration during metastasis[11–13]. Hence, podosome formation and regulation is critical to disease pathophysiology and homeostasis.

Akin to the widely studied focal adhesions (FAs), podosomes have been shown to exert mechanical forces and to respond to ECM stiffness[14–20]. Whereas FAs assemble into fibrillar microscale structures that apply contractile forces to the substrate[14,21–25], podosomes assemble into a columnar architecture consisting of an actin core surrounded by a ring complex containing adhesome proteins including integrin receptors[26]. The actin core protrudes from the cell body, applying nN compressive forces onto the underlying substrate[16,27,28]. Given that a static cell cannot experience a force imbalance[29], it is widely recognized that podosomes apply opposing tensile forces, with some disagreement as to the requirement for integrin adhesion forces[26–28,30–33]. Mathematical modeling suggests that these tensile forces are localized to the podosome ring[27], and there are two lines of experiments that support this model. The first comes from biophysical measurements of talin extension[18] and vinculin tension[28] within podosomes. These measurements are indirect as they fail to map the molecular forces applied by the podosome itself. The second class of measurements reports bulk substrate deformations using traction force microscopy (TFM). While TFM provides a direct measure of cellular stresses, the spatial and force resolution of the method precludes mapping the forces at the podosome ring complex. A more sensitive variant of TFM that is interferometry based offers improved force sensitivity but still averages deformations of the substrate[34] and thus cannot quantify receptor forces. To the best of our knowledge, no quantitative force maps have been reported validating the role of adhesion receptor mechanics in opposing actin protrusion and mechanically linking the substrate and the cytoskeleton within podosomes.

Further confounding podosome mechanical models, recent results demonstrated the formation of podosome-like adhesions on supported lipid bilayers (SLBs)[35,36]. SLBs are phospholipid membranes that self-assemble onto a glass slide. Lipids are confined in the z-direction but are laterally fluid[37]. Thus, on SLBs, podosomes are reported to form even in the absence of traction forces[35,38], which is confounding since podosomes apply compressive forces on the SLB.

In this work, we employ DNA-based mechanotechnology tools to challenge the hypothesis that integrins cannot apply forces on fluid membranes and to investigate the role of integrin tensile forces in podosome mechanosensing. We recently coined the term DNA mechanotechnology to refer to nucleic acids that sense, generate, and transmit forces.[39] As a material, DNA offers the ability to map and perturb receptor forces with piconewton (pN) force resolution and submicron spatial resolution. We first quantify podosome-mediated depletion as a marker of actin core protrusion on SLBs. Next, we introduce molecular tension-fluorescence lifetime imaging microscopy (MT-FLIM), producing pN maps of integrin tension during receptor–ligand clustering at the cell membrane. Previously, molecular tension-fluorescence microscopy (MTFM) imaging on supported bilayers was carried out using ratiometric probes[40,41], but these induced artificial clustering or employed three-way energy transfer, which hinder quantitative analysis. Moreover, we found that conventional MTFM probes[23] are static quenched and poorly suited for FLIM. Thus, we report a re-engineered DNA-based FLIM–FRET probe that circumvents these problems. To better understand podosome tensile forces, we apply a recently developed force-orientation analysis technique to demonstrate that integrin forces are primarily perpendicular to the substrate. Treatment with pharmacological inhibitors showed that podosome tensile forces are a direct consequence of actin polymerization.

The latter section of our paper focuses on employing DNA mechanotechnology tools to manipulate podosome forces and reveal mechanoregulation. Specifically, using photocleavable (PC) DNA, we introduced local mechanical defects within podosomes with subcellular resolution, allowing us to track changes in tension and protrusion. These experiments demonstrate that podosome–podosome mechanical crosstalk is minimal within the cell models studied and that force balance is controlled locally. With tension-gauge tethers (TGTs), which limit the maximum magnitude of integrin tension, we demonstrate that podosomes not only apply integrin forces in their rings, but that these forces are required to stabilize podosomes.

Finally, we model pN integrin forces in podosomes and demonstrate that podosomes exert nN vertical forces on an SLB, in agreement with previously measured protrusion forces[16]. Our work offers receptor-level quantitative maps of integrin tension on fluid substrates and provides direct experimental evidence that podosomes are mechanosensors with local pN sensitivity.

## Results

**Podosome actin content correlates with RGD-probe depletion.** Cyclic Arg-Gly-Asp-D-Phe-Lys (cRGDfK, RGD) peptides were tethered to the SLB through DNA oligonucleotides (Fig. 1a, Supplementary Tables 1 and 2, Supplementary Figs. 1–4). RGD-functionalized SLBs displayed a lateral diffusion coefficient of $1.41 \pm 0.07\,\mu m^2\,s^{-1}$ (Fig. 1b)[42]. Within tens of minutes of culturing NIH 3T3 fibroblasts onto these substrates, we observed the formation of podosome-like adhesions[35,36]. These adhesions were actin rich, excluded the RGD–DNA ligand (Fig. 1c) and were surrounded by vinculin and phospho-paxillin (Supplementary Fig. 4). Since these podosome-like adhesions contained the key elements of podosomes[35], we refer to these structures as podosomes throughout this manuscript. The depletion of RGD–DNA was caused by pushing forces generated by the protrusive actin core[35]. Note that these forces exclude RGD–DNA but do not rupture the underlying SLB (Supplementary Fig. 5), which is largely incompressible[37,43,44]. The f-actin rich core of podosomes was anti-localized with RGD–DNA (Fig. 1d), and increased actin content correlated with RGD depletion (Fig. 1e). Podosomes that depleted more RGD (measured by % decrease in fluorescence) also had larger depletion radii (Fig. 1f). Given these data and the published finding that actin content and protrusion forces display similar behavior[28,45], we reasoned that RGD depletion serves as a suitable proxy for podosome protrusive forces on an SLB.

**MT-FLIM maps pN receptor tension and clustering on SLBs.** To determine whether integrins apply tensile forces in podosomes on an SLB, we developed MT-FLIM to visualize receptor forces applied on fluid membranes. MTFM has permitted mapping of receptor forces on rigid substrates[22–24,46–51] but is challenging to apply on fluid interfaces such as SLBs, because tension signal and probe density are convolved as receptors cluster[37]. Two ratiometric approaches have aimed to address this problem. The first used 15 nm particles tagged with MTFM probes and a reference dye,[40] but this leads to artificial clustering of receptors as they

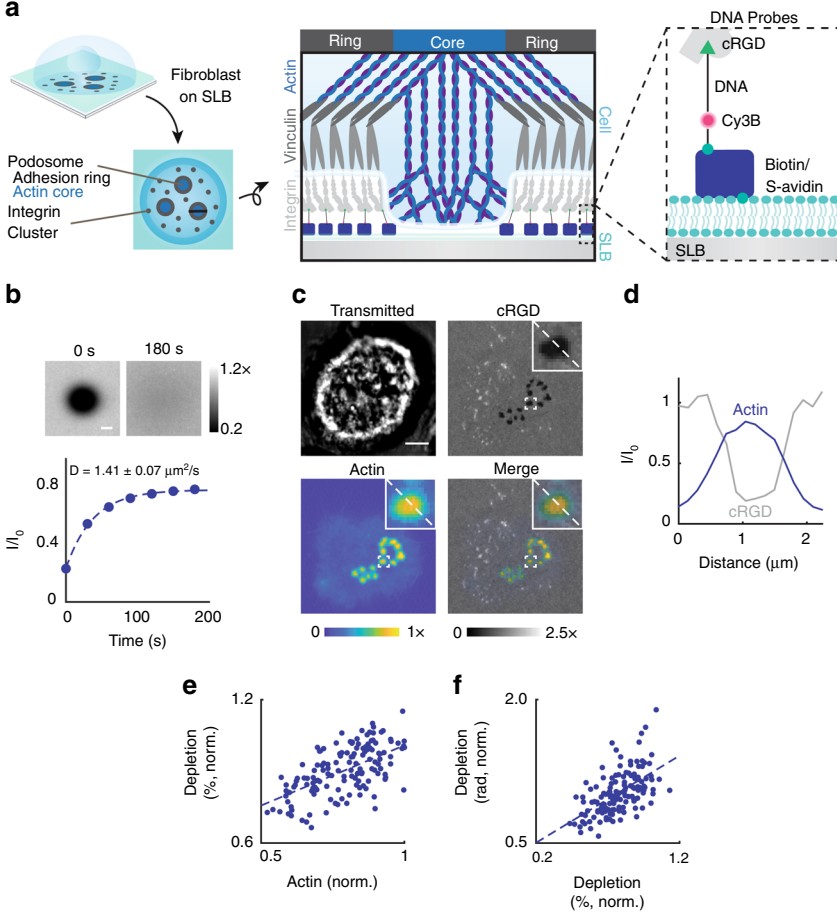

**Fig. 1** NIH 3T3 fibroblasts form protrusive podosomes on SLBs. **a** (Left) Schematic of the SLB podosome model. (Middle) Schematic of a single podosome at the cell–SLB interface across the black line. Podosomes exclude RGD-probes at their protrusive core. (Right) Zoomed-in schematic of a single DNA probe. **b** Representative images and mean intensity plot of RGD–DNA probes in a fluorescence recovery after photobleaching experiment. Data were normalized to the SLB intensity at $t = 0$ s. DNA probes had a diffusion coefficient of $1.41 \pm 0.07\ \mu m^2\ s^{-1}$ (mean + s.e.m., $n = 9$ FRAPs, 3 experiments). Scale bar, 10 μm. **c** Representative podosome-forming cell on an SLB decorated with single-stranded DNA probes. Cells were transfected LifeAct to visualize actin cores. Scale bar, 5 μm. **d** Linescan analysis across the zoomed-in podosome in **c**. cRGD and actin intensity are normalized to the SLB background and the brightest pixel, respectively. **e**, **f** Scatter plots of actin intensity versus depletion (%) and depletion (%) versus depletion radius. All data were normalized by the brightest podosome per cell to account for varying LifeAct expression. ($r = 0.657$, 0.604, $N = 161$ podosomes, 23 cells, three experiments). Source data are provided as a Source Data file

bind to the multivalent ligands presented by the nanoparticle, which may modulate integrin signaling[52,53]. The latter method leverages DNA-based MTFM probes tagged with three organic dyes[41]; however this probe likely experiences difficulty to correct through-space energy transfer pathways between the three chromophores.

We hypothesized that FLIM could be used to map receptor forces on fluid membranes, because fluorescence lifetime is sensitive to energy transfer but is independent of dye concentration[54]. We functionalized SLBs with FRET-based DNA tension probes containing binary DNA hairpins with a tunable $F_{1/2}$ threshold and two linker arms; the $F_{1/2}$ threshold is defined as the force equilibrium at which 50% probes open (Fig. 2)[23]. The bottom arm was hybridized to a biotinylated quencher strand containing an internal deoxythymidine BHQ1 modification. We selected this site for the quencher to ensure that the probe was FRET quenched, as absorbance spectroscopy demonstrated that conventional MTFM probes are static quenched and thus poorly suited for FLIM (Supplementary Fig. 6). The upper arm of the DNA hairpin was hybridized to a Cy3B ligand strand containing cRGD. At rest, closed probes are FRET quenched, with a low fluorescence intensity and a short fluorescence lifetime. When

integrin receptors bind and transport DNA tension probes into nascent integrin adhesions, the probes are clustered, causing an increase in fluorescence intensity. If the applied force equals or exceeds $F_{1/2}$, then the DNA hairpin unfolds, causing an increase in both fluorescence intensity and fluorescence lifetime (Fig. 2a).

Because of the limited photon-budget[55,56] and the probes' multiexponential fluorescence decay (Supplementary Table 3, Supplementary Fig. 7a–c), we resorted to fit-free FLIM imaging in which the average lifetime is reported as the barycenter of photon arrival time. By this definition of average lifetime, multiexponential decays do not exhibit a linear relationship between average fluorescence lifetime and FRET efficiency[57,58]. Therefore, we generated empirical calibration curves to relate the percentage of opened DNA probes to the measured fluorescence lifetime (Fig. 2b). This was achieved by imaging a small library of SLBs presenting a mixture of opened and closed DNA probes (Supplementary Fig. 7d, e). We also characterized these SLBs using epifluorescence to determine probe quenching efficiency (QE) (Fig. 2c) and as a control, we measured intensity and fluorescence lifetime as a function of probe density (Supplementary Fig. 8). We found a subtle decrease in probe lifetime with increasing probe density.

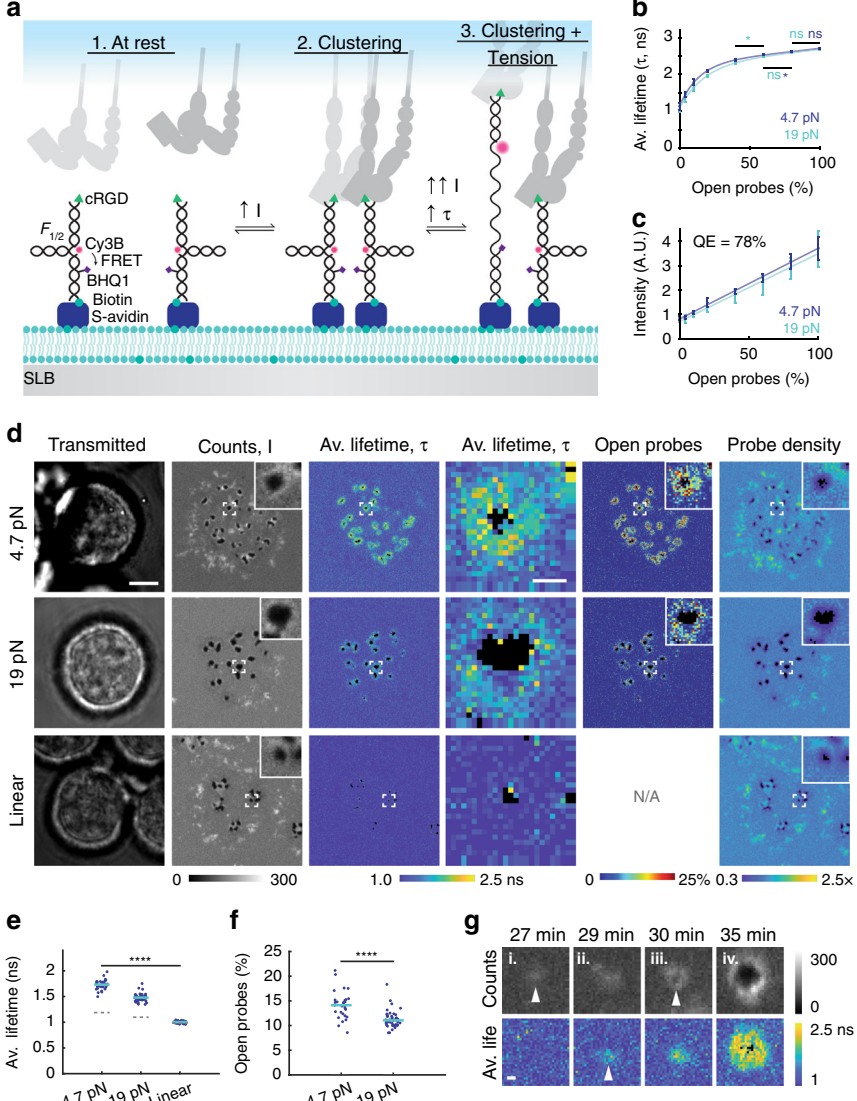

**Fig. 2** Fibroblasts exert pN integrin tension in podosome rings on SLBs. **a** MT-FLIM probes report clustering and tension. 1. In closed probes, donor fluorescence (Cy3B) is FRET quenched by BHQ1 quencher. 2. Receptor binding and clustering increases probe density and local intensity. 3. When integrin receptors apply tension above $F_{1/2}$, MT-FLIM probes unfold, yielding a massive increase in both fluorescence lifetime and fluorescence intensity. **b** Average Cy3B fluorescence lifetime of 4.7 (dark blue line) and 19 (teal line) pN tension probes increases with increasing percent open probes. Fits ($r^2 = 0.999$) were used to determine the percentage of probes opened by cells. Data represent the mean fluorescence lifetime from Gaussian fitting of histograms ± s.e.m. (error bars) across 2–3 SLBs per data point. **c** MT-FLIM probe intensity of 4.7 (dark blue line) and 19 (teal line) pN tension probes increases linearly with open probe fraction. Probes are 78% quenched when folded ($r^2 = 0.995$). Data represents the mean ± s.e.m. (error bars) across at least three SLBs. **d** Representative images of cells forming podosomes on linear, 4.7 pN, and 19 pN MT-FLIM probes. Black pixels in the Av. Lifetime channel indicate pixels with <25 photons or $\tau > 2.97$ ns. Scale bar, 5 µm, zoom-in scale bar, 0.5 µm. **e** Average podosome fluorescence lifetime per cell on linear, 4.7 pN, and 19 pN MT-FLIM probes. Teal solid lines represent the mean. Dashed gray lines represent average SLB background fluorescence lifetime. At least 42 cells were analyzed per condition, 3–4 experiments per condition. Statistics were performed with a Kruskal–Wallis ANOVA. **f** "Mean per cell" percent open probes in podosomes. Teal lines represent the median. Statistics were performed using a rank sum test. ****$P < 0.0001$. **g** MT-FLIM images showing the emergence of i. clustering, ii. tension, and iii. protrusion. The mature podosome is shown in iv. Each frame is centered on the podosome or cluster centroid. White triangles indicate the emergence of clustering, tension, and depletion. Scale bar, 0.3 µm. Source data are provided as a Source Data file

However, this effect was minor compared with the effect of force-mediated hairpin opening.

To map the podosome tensile landscape, we imaged $F_{1/2} = 4.7$ pN tension probe signal generated by podosome-forming fibroblasts on SLBs (Fig. 2d). In the photon counts channel, we observed three populations of signal: negative signal corresponding to actin-mediated depletion at the podosome core, bright rings that surrounded the depletion zones, and bright puncta distributed across the cell–SLB junction, which we termed integrin clusters. Both clusters and podosome rings colocalized

with β1 integrin, confirming that they were caused by integrin-mediated adhesions (Supplementary Fig. 9). The fluorescence lifetime increased in podosomes, indicating that integrins mechanically unfolded DNA hairpins with $F \geq 4.7$ pN. In contrast, the bright puncta outside of podosomes did not show changes in lifetime; thus integrin forces were <4.7 pN in these regions. Using our calibration curves combined with the average lifetime maps, we calculated the percentage of opened probes. By combining these data with a mathematical framework, we also extracted the local probe density (Supplementary Note 1). To

understand the limits of detection in low density regions, we analyzed MT-FLIM photon statistics (Supplementary Note 2, Supplementary Fig. 10). Note that low photon count pixels, such as within depletion zones, produce less reliable lifetime values; these pixels were excluded according to lifetime and photon count cutoffs described in Supplementary Note 2 (Supplementary Fig. 11). Images represent the accumulation of signal over 1 min, so there is some spatiotemporal averaging. Nevertheless, these images provide the first quantitative maps of integrin forces on an SLB and within podosomes. Moreover, our conclusions are generalizable: MT-FLIM imaging of Src-transformed mouse embryonic fibroblasts indicated that this spatial distribution of forces is also common in invadosomes and invadosome belts (Supplementary Fig. 12).

We next challenged podosomes using a more stable hairpin with $F_{1/2} = 19$ pN. Fluorescence lifetime increased in podosome rings but corresponded to a lower percentage of open probes. This suggested that a smaller subset of integrins apply $F \geq 19$ pN compared with $F \geq 4.7$ pN (Fig. 2e, f). Control experiments employing a linear DNA duplex lacking a hairpin secondary structure exhibited nearly identical photon counts signal as the hairpin probes, showing regions of depletion and bright puncta. These regions did not exhibit significant changes in fluorescence lifetime. In addition, control experiments in which the fluorophore–quencher pair and ligand were separated onto two independent co-presented DNA probes showed no change in fluorescence lifetime (Supplementary Fig. 13). Together, these data confirm that changes in lifetime are due to integrin-mediated mechanical unfolding of the hairpin.

To investigate the time course of podosome mechanical force generation, we next collected time-lapse MT-FLIM images of cell spreading and podosome formation (Fig. 2g, Supplementary Fig. 14, Supplementary Movie 1). Within 20 min of plating, cells formed dynamic integrin clusters, which lacked detectable forces. Clustering was followed by tension and podosome-mediated depletion. Depletion was either preceded by tension (as shown) or occurred simultaneously with tension. Note that resolving the order of events with higher time resolution was difficult to achieve because of the 60 s FLIM acquisition time. Nonetheless, we found that integrin tension increased as podosomes became larger and more protrusive. Podosomes tended to move outward toward the cell periphery, and clusters grew by de novo podosome formation at common nucleation sites or by splitting (Supplementary Fig. 14). These time-resolved tension measurements show tight spatiotemporal coordination between podosome protrusive forces and tensile forces at the single podosome level.

**Podosome forces are primarily vertical.** Given that SLBs cannot support lateral traction forces and that actin cores exert pushing forces on the substrate, we hypothesized that the integrin forces in podosome rings would be primarily oriented in the vertical direction. Alternatively, the protrusive podosome core could serve as a diffusion barrier[37,59] to support podosome lateral forces. To test this, we used molecular force microscopy (MFM)[49], a recently developed fluorescence polarization-based method to quantify the direction of receptor forces (Fig. 3a–d). Whereas MT-FLIM reports the magnitude of tension, MFM maps force orientation. DNA-based tension probes reorient in the direction of applied integrin forces. In MFM, cyanine fluorophores are rigidly coupled to the DNA duplex[60]. Thus, integrin forces dictate both the orientation of the DNA tension probe and of its attached fluorophore. MFM utilizes excitation resolved polarization microscopy to measure the orientation of the fluorophore in order to deduce the strained-DNA orientation. MFM leverages mechano-selection, in which only open probes contribute significantly to

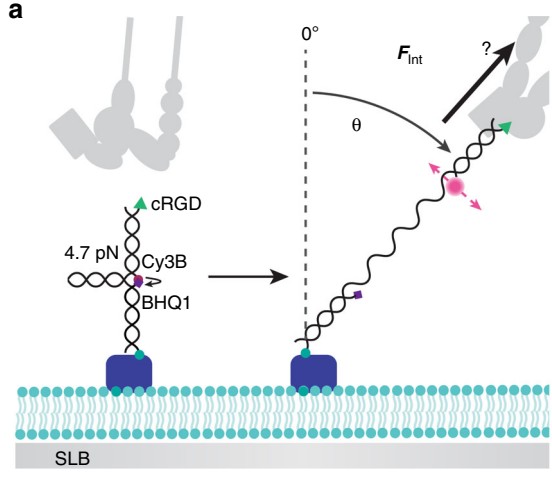

**a**

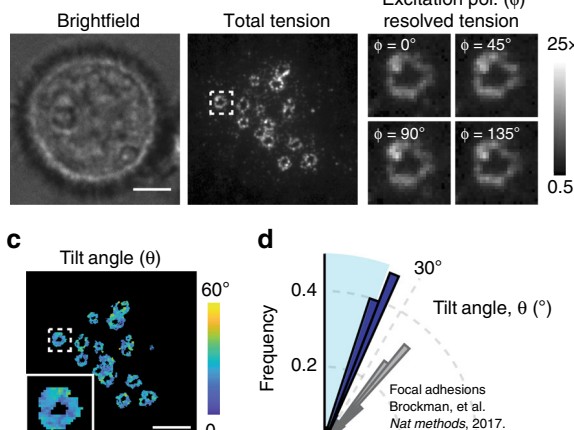

**b**   Brightfield   Total tension   Excitation pol. (φ) resolved tension
φ = 0°   φ = 45°   25×
φ = 90°   φ = 135°   0.5

**c** Tilt angle (θ)   60°   **d**   30°   Tilt angle, θ (°)   Focal adhesions Brockman, et al. *Nat methods*, 2017.   90°

**Fig. 3** Integrins exert vertical forces in podosome rings. **a** Schematic of MFM on an SLB. When an integrin receptor binds and applies forces above 4.7 pN, the probe unfolds and generates Cy3B fluorescence. Probes orient along the applied force vector, allowing determination of the tilt angle, θ, from the excitation polarization dependent Cy3B fluorescence. **b** Representative podosome MFM data after ~90 min cell spreading. Total tension is the maximum tension signal per pixel. Podosome zoom-ins depict the normalized Cy3B fluorescence at four different excitation polarizations. Scale bar, 5 μm. **c** Computed tilt angle map for the cell shown in **b**. Scale bar, 5 μm. **d** Angular histogram of average podosome tilt angle per cell, θ. Vertical forces are indistinguishable within the ~20° cone, represented by the blue region (N = 25, three experiments). Source data are provided as a Source Data file

fluorescence and polarization signal, thereby suppressing the contribution of nonoriented closed probes. Accordingly, we substituted MT-FLIM probes for static quenched tension probes, in which the Cy3B fluorophore and BHQ1 quencher are in direct contact, thus offering mechano-selection of the Cy3B signal (Fig. 3a, Supplementary Fig. 15a, b). Although these probes lack a density reporter, they provide a suitable proxy for integrin forces in this system, because podosome rings tended to show only minor enrichment in the MT-FLIM density channel (Fig. 2d). We confirmed that the effects of clustering and dynamics were minor using linear MTFM probes (Supplementary Fig. 15c–g). MFM fluorophore orientation measurements were validated using a membrane-bead standard as recently described[49] (Supplementary Fig. 16).

MFM analysis of podosome rings suggested that forces were primarily vertical or disorganized (Fig. 3b–d, Supplementary

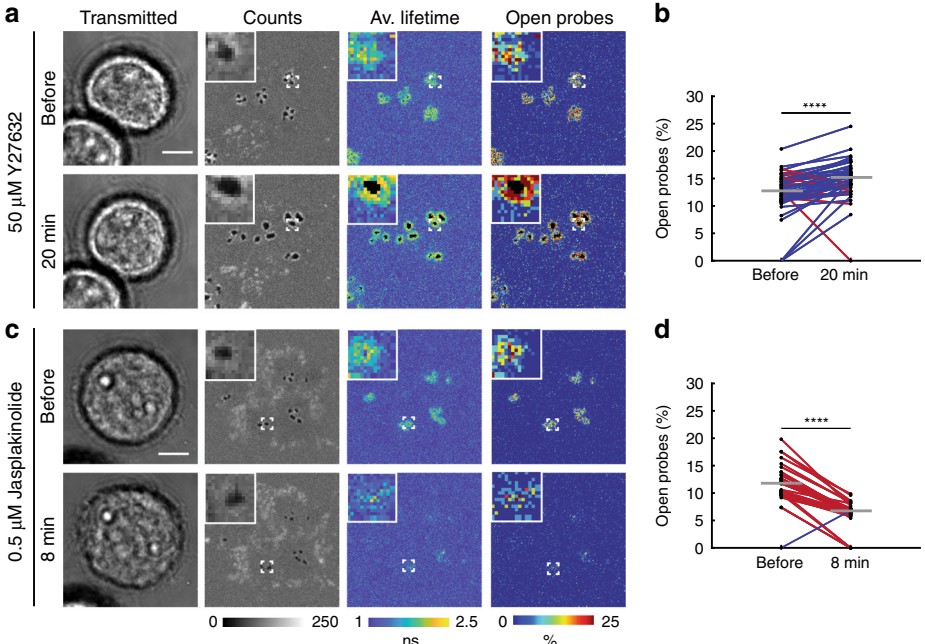

**Fig. 4** Actin polymerization drives integrin tension. **a**, **c** Representative before and after MT-FLIM images of NIH 3T3 fibroblasts on 4.7 pN MT-FLIM probes treated with 50 μM Y27632 or 0.5 μM jasplakinolide, respectively. **b**, **d** Average percent open probes in podosomes per cell before and after drug treatment. Blue and red lines represent an increase or decrease in percent open probes per cell, respectively. Gray horizontal lines represent the median percent open probes. Statistics were performed with a two-tailed Wilcoxon matched-pairs signed rank test. At least 34 cells from three experiments were analyzed per condition. ****$P < 0.0001$. All scale bars, 5 μm. Source data are provided as a Source Data file

Fig. 16g). The average podosome tilt angle per cell was 21° ± 2° from vertical. In comparison, integrins in fibroblast FAs on glass exert forces with a tilt angle of 40° ± 2°[49]. MFM has limited sensitivity to tilt angles less than 20° from the normal and becomes more susceptible to noise for these small tilt angles. Therefore, these MFM results are consistent with podosome tension that is primary oriented normal to the SLB substrate[49].

To support our MFM result, we performed emission-resolved polarization measurements[49] (Supplementary Fig. 17). Here, any global organization in the lateral component of podosome forces within the ring will lead to bright nodes in fluorescence anisotropy. In agreement with a model of vertical force generation, we found no organized pattern in podosome fluorescence anisotropy. This confirms MFM measurements, indicating that podosomes on SLBs lack an organized traction force component in the plane of the SLB.

**Integrin forces are actin but not myosin IIa dependent**. To understand the mechanism of integrin force generation in podosomes, we treated cells with a panel of small-molecule inhibitors and quantified changes in 4.7 pN tension and core depletion size using MT-FLIM. We first treated cells with Y27632 to inhibit Rho kinase and found that integrin forces were not diminished (Fig. 4a, b). Surprisingly, we observed an increase in the percentage of open probes and core depletion size (Supplementary Fig. 18a). To test whether this observation was caused by myosin II inhibition, we treated cells with blebbistatin (Supplementary Fig. 19a–c). Although it was reported that blebbistatin releases tension on genetically encoded vinculin tension sensors in macrophages on immobile ligands[26], we observed only a slight change in integrin tension ($p = 0.0464$). Podosome core size was not affected ($p = 0.3603$). We further validated this result by knocking down *MYH9*, the myosin IIa head domain. No significant changes in podosome depletion or tension were observed when *MYH9* was knocked down (Supplementary Fig. 19d–h).

Because actomyosin contractility was largely dispensable in generating tension in integrin receptors on SLBs, we next treated cells with jasplakinolide, which is a potent actin stabilizer and polymerizer that causes reorganization of the cytoskeleton into disorganized aggregates[61,62]. Following 500 nM jasplakinolide treatment for 8 min, core depletion area decreased, and integrin tension in podosome rings was reduced (Fig. 4c, d, Supplementary Fig. 18b). These data support a model in which actin polymerization in the podosome alone is sufficient to support integrin-based tension in the podosome ring complex.

**Protrusion and tension engage in a mechanical feedback loop**. We next sought to perturb podosome tension to determine how podosomes respond to external mechanical inputs (Fig. 5a). To achieve this goal, we engineered releasable MTFM probes by anchoring probes with a PC biotin group (Fig. 5a, b). This PC modification allows for optical manipulation of integrin tension with high spatial and temporal resolution (Supplementary Fig. 20). Cells were cultured on SLBs with PC probes for ~1 h. Then, integrin ligands anchoring podosomes were severed with a 405 nm laser (Fig. 5c). We anticipated that optical release of DNA probes would terminate integrin tension and cause rapid refolding of the DNA hairpin and re-quenching of tension signal in the podosome ring (Supplementary Fig. 21). Upon 405 nm illumination of a 7 μm² podosome-containing region, we tracked the changes in tension signal both at the site of photocleavage (proximal) and across the entire cell (distal) (Supplementary Fig. 22). Although SLBs dissipate long-range forces, we still wondered if it would be possible to observe global changes that were communicated intracellularly, such as through altered diffusion of adhesion proteins[28]. In severed podosomes proximal to photocleavage, we observed a massive loss of tension signal followed by a gradual remounting of tension (Supplementary Fig. 21), which can be attributed to the replenishing of surface-bound ligands by diffusion. In contrast, podosomes distal to the

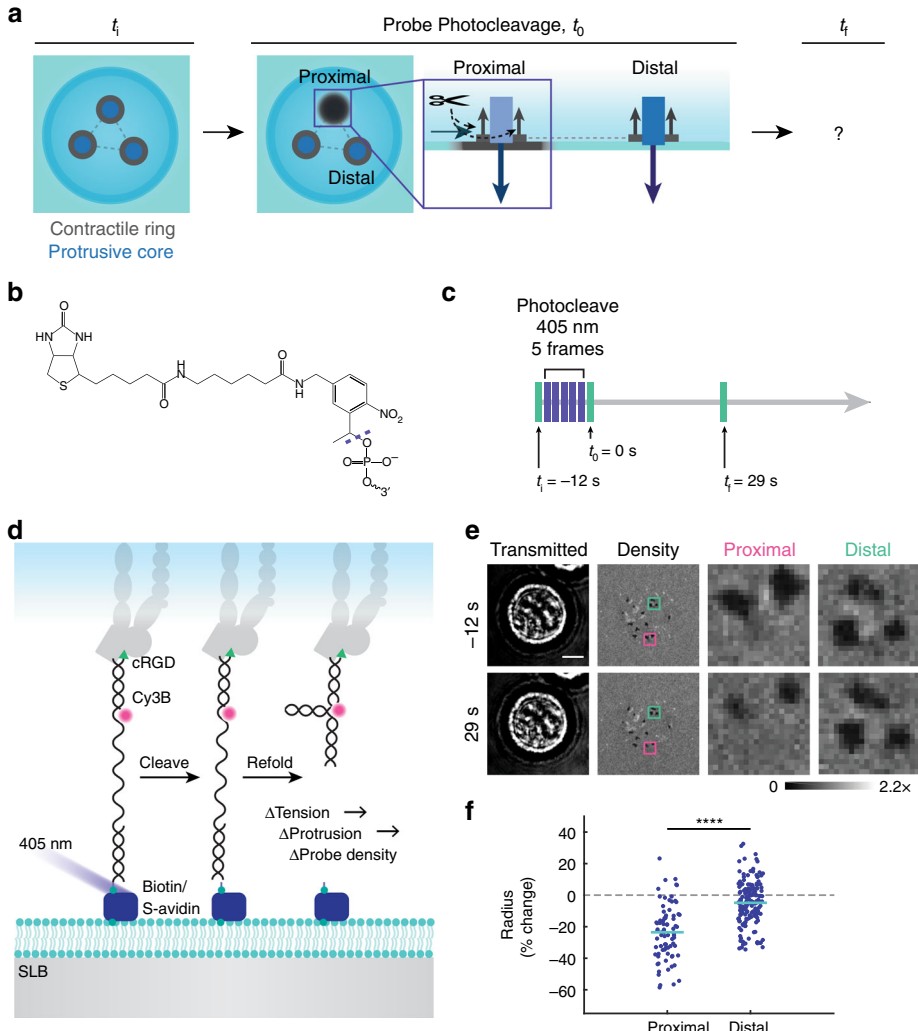

**Fig. 5** Loss of integrin ring tension causes local podosome retraction. **a** To test how the podosome network responds to perturbations in podosome tension, probes were photocleaved under individual podosomes or podosome clusters, and the proximal and distal responses were monitored. **b** Chemical structure of PC Biotin, which was incorporated at the 5′ termini of 4.7 pN hairpins. **c** Cells were imaged before and after five sequential frames of probe photocleavage with a 405 nm laser. **d** Schematic of photocleavable protrusion probes. When the biotin anchor is cleaved with a 405 nm laser, the probe detaches from the bilayer and refolds, severing the podosome's mechanical connection to the SLB. Cy3B intensity reports probe density, and therefore serves as a marker of core size. **e** Representative images of cells and podosomes before (−12 s) and after (29 s) photocleavage. Proximal pink boxes indicate the entire region of photocleavage. Distal green boxes show a representative region across the cell. **f** Summary statistics showing the percent change in podosome radius as measured by Cy3B depletion in regions cleaved by the 405 nm laser (proximal) and all other regions of the cell (distal). Statistics were performed using a two-way ANOVA (control groups in Supplementary Fig. 23). At least 79 podosomes were analyzed per condition, three experiments. Outlier podosomes (median ± 3 median absolute deviations) were excluded. ****$P < 0.0001$. Scale Bar, 5 μm. Source data are provided as a Source Data file

site of photocleavage exhibited only a small reduction in tension signal.

To quantify the changes in podosome protrusiveness upon loss of integrin tension, we synthesized PC probes lacking the quencher (Fig. 5d). Here, Cy3B intensity is insensitive to tension, and changes in signal exclusively report DNA density. Hence, probe depletion radius provided a proxy for the change in protrusive forces applied by the podosome's actin core. We tracked individual podosomes before (−12 s) and after (29 s) photocleavage and found that core depletion radius was reduced proximal to photocleavage (−23.5 ± 17.0%). Distal to ablation, podosome radius was slightly reduced (mean = −4.9 ± 13.8% change) (Fig. 5e, f, Supplementary Fig. 23). Control experiments confirm that mechanical perturbations were largely responsible for the observed changes in signal, with phototoxicity contributing minimally at the 29 s time point (Supplementary Figs. 21 and

23). Note that at later time points, some cells do show some light-mediated podosome disassembly. These results confirm local mechanical feedback between integrin tension and actin protrusion as the primary mechanism of force balance in podosomes on SLBs in this 3T3 cell line and suggest that there may be a minor contribution of mechanical coupling across the podosome network. Other podosome-forming cell types such as myeloid cells may display different levels of podosome–podosome coupling.

**Efficient podosome formation requires integrin tension.** Given podosomes' tendency to shrink following loss of tension, we hypothesized that integrin forces are not only important for podosome maintenance but also for podosome initial formation and maturation. To test the hypothesis that integrin

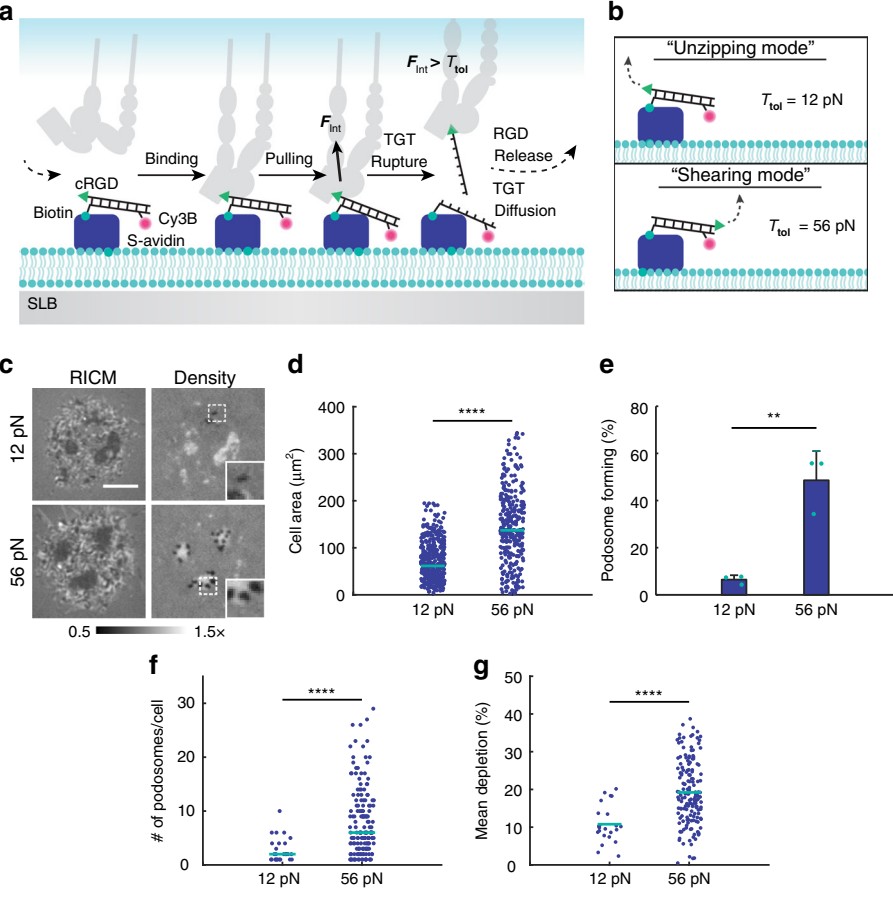

**Fig. 6** Podosomes maturation requires $F > 12$ pN integrin tension. **a** Schematic depicting integrin-TGT interaction on an SLB. First, an integrin receptor binds to a TGT duplex on the SLB. Upon $F_{Int} > T_{tol}$, the TGT irreversibly ruptures, creating a transient local concentration gradient. Probe density is replenished through diffusion, and the cycle repeats. **b** $T_{tol}$ is set by relative orientation of the cRGD ligand and the biotin anchor. **c** Representative images of podosomes on TGTs. Scale bar, 5 µm. **d** Cell area on TGTs. Teal bars represent the median area. Statistics were performed with a rank sum test. At least 332 cells were analyzed per condition, three experiments. **e** Percentage of cells forming podosomes on TGTs. Bars represent the mean ± s.d. (error bars), and teal circles represent individual experiments. Samples were compared using an unpaired two-tailed Student's $t$ test. At least 76 cells were analyzed per surface, three experiments. **f** Scatter plot showing the number of podosomes per cell in podosome-forming cells on 12 versus 56 pN TGTs. Teal bars represent the median. Statistics were performed using a rank sum test. At least 25 cells were analyzed per condition, three experiments. Outliers more than three quartiles above or below the upper quartile were eliminated. **g** Scatter plot showing the mean podosome depletion on 12 versus 56 pN TGTs. Teal bars represent the mean depletion. Statistics were performed using a two-tailed unpaired Students $t$ test. At least 25 cells were analyzed per condition, three experiments. Cells spread for 2 h on TGTs prior to imaging. $**P < 0.01$, $****P < 0.0001$. Source data are provided as a Source Data file

tension is required for the efficient formation of podosomes, we employed TGTs, which limit the magnitude of integrin forces[63] (Fig. 6). Here, TGT probes consist of a Cy3B-labeled DNA duplex that is anchored to the SLB. Upon the application of integrin forces that exceed the tension tolerance ($T_{tol}$, defined as the threshold force leading to mechanical melting at the 2 s time scale) of the probe, the duplex irreversibly ruptures. Thus, TGTs limit the maximum tension per receptor (Fig. 6a). The $T_{tol}$ value can be tuned by changing probe geometry. In the unzipping mode, the cRGD ligand and biotin are on the same terminus of the duplex, and $T_{tol} = 12$ pN. In the shearing mode the biotin anchor and the cRGD ligand are on opposite sides of the duplex, and $T_{tol} = 56$ pN (Fig. 6b). TGT experiments are unique in that they present chemically identical substrates that differ in their molecular stiffness. Thus, these experiments could also be used to test whether mechanosensory podosomes have pN sensitivity. We hypothesized that podosomes would be sensitive to these pN changes in ligand stiffness and that 12 pN TGTs would hinder podosome formation.

On 56 pN TGTs, cells exhibited increased spreading area, and eightfold more cells formed podosomes compared with SLBs with 12 pN TGTs (Fig. 6c–e). Within podosome-forming cells, the number of podosomes per cell and the mean depletion in podosomes were significantly reduced on 12 pN TGTs (Fig. 6f, g). These data reveal that limiting integrin tension to a value of 12 pN produces weaker protrusive forces. Time-lapse imaging of cells cultured on TGT substrates revealed that cells formed podosome-mediated depletions regardless of $T_{tol}$, but podosomes tended to be smaller and more transient on 12 pN TGTs compared with on 56 pN TGTs (Supplementary Movies 2 and 3). Thus, podosome growth and stability is directly regulated by integrin forces.

## Discussion

Although previous works have suggested that podosomes function as independent mechanosensors, the nature of the mechanical linkage between the substrate and the cytoskeleton was unknown[16,19,27,30,64]. Our work demonstrates that podosome

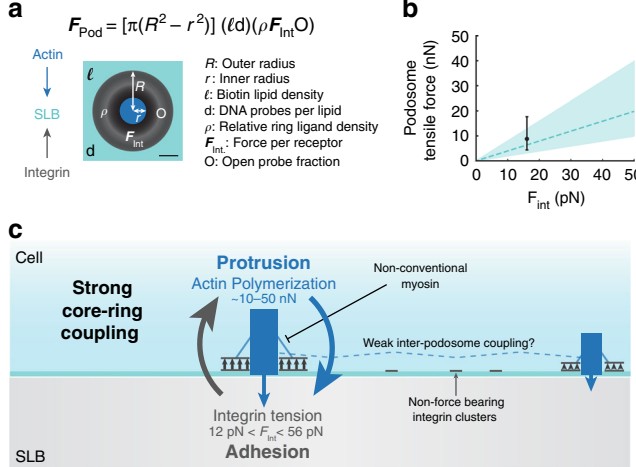

**Fig. 7** Podosome ring modeling estimates nN tensile forces. **a** Schematic and equation for modeled podosome tensile forces on an SLB. Modeled podosome consist of a 1 μm outer radius and a 0.3 μm inner radius. The actin core exerts protrusion forces on the SLB, and the integrin receptors in the adhesion ring apply tensile forces. Modeled podosomes were parameterized using data from Fig. 2 and Supplementary Fig. 18. Scale bar, 2.5 μm. **b** Plot of ring tensile forces as a function of the per receptor force, $F_{Int}$. The dashed line indicates total tensile forces assuming 1 probe per biotinylated lipid, and the black data point represents the minimum force predicted by MT-FLIM. Confidence intervals and error bars correspond to 0.5 to 2 probes per biotinylated lipid. **c** Model of podosome forces on an SLB. Individual podosomes exert protrusion forces on the SLB with the actin core, while integrin receptors tug on RGD ligands in the ring domain. Podosomes experience strong mechanical coupling between protrusive and tensile forces. This coupling is focused within a single podosome and is weak in-between podosomes. This equilibrium can be shifted through myosin or actin inhibition. Podosomes with more actin content have smaller depletion areas and exert less tension. Receptor clusters that are not associated with podosomes do not experience detectable forces. Source data are provided as a Source Data file

integrin receptors apply pN vertical tension to their ligands, and that these tensile forces are required for actin core polymerization. This is a significant departure from past works, which have used the SLB model as evidence that podosome formation occurs in the absence of adhesion forces and have stated that integrins cannot apply forces to ligands on fluid SLBs[35,65,66]. Instead we observed the formation of integrin clusters[65], followed by core and ring growth that coincided with at least 19 pN integrin tension in the ring. Clustered integrins not associated with podosomes on SLBs did not apply detectable forces. To contextualize podosome forces on SLBs, we used our data to parameterize a simple adhesion model (Fig. 7a, b). Each individual podosome applied nN vertical tensile forces, which was the sum of pN integrin tension and was strikingly similar to the protrusion forces measured for podosomes on FORMVAR, which is nonfluid[16].

A key question in the podosome literature is whether podosome tensile forces are generated directly from core polymerization or whether actomyosin contractility is required to generate tension in the ring[27,28,31,32]. Our work validates a model (Fig. 7c) of polymerization-induced tension on the ring, as integrin tension was abolished when actin polymerization was perturbed. Few works have reported podosome formation on substrates lacking ligand[67,68], leading to a hypothesis that the podosome architecture can sustain tension without receptor forces[19]. Our TGT data disagree with this model, demonstrating that firm integrin adhesion is required to form stable and protrusive podosomes. In

contrast to works performed on rigid substrates[28,69–71], our data support a model of local podosome force balance that is independent of myosin IIa contractility. Van den Dries et al. reported that myosin IIa inhibition led to increased actin core intensity by shifting the feedback between contractility and protrusion[28]. We observed a similar increase in core depletion area and tension signal with Rho kinase inhibitor. It will be important to identify which myosins plays a role in this vertical force balance feedback on an SLB. An interesting candidate is myosin 1, which has been shown to localize to podosome cores and to apply forces when anchored to a fluid membrane[72–74].

Our results should be considered in the context of a few important caveats. While fibroblasts provide a robust model to study podosomes in vitro, to our knowledge, fibroblasts have not been shown to form podosomes in vivo. Another corollary point is that the mechanical properties and the stiffness of the substrate will likely influence podosome dynamics. While MT-FLIM quantitatively maps integrin tension on fluid substrates, this signal is subject to some spatiotemporal convolution, and fast changes in tension will be obscured during the 1-min acquisition. Furthermore, regions of low photon counts such as podosome depletion regions may be disproportionately affected by the diffraction limit. These issues will be addressed as superresolved and faster FLIM electronics become more widely available.

Nevertheless, our work provides valuable mechanobiology tools and insight. SLBs offer a unique landscape to study the minimal mechanical machinery required for podosome formation, and DNA probes provide a powerful method to map and manipulate molecular forces. While RGD ligands on an SLB are more mobile than in physiological ECM, their mobility can recapitulate degraded ECM[75], which is relevant to podosome and invadosome biology[1,13]. During cancer cell invasion, cells alternate between periods of integrin and invadosome-mediated matrix degradation and migration[30,76]. Thus, podosome retraction following adhesion photocleavage may provide a model to understand how changes in tension can regulate function. While MT-FLIM has some current limitations in its temporal resolution, this method offers a unique solution to mapping forces at fluid interfaces and is a departure from past methods in its incorporation of a parameter that uniquely reports forces and is not subject to intramolecular fluorescence crosstalk. PC probes provide a method to perturb individual adhesion structures with minimal disruption to the cell body and without changing the extracellular environment[77]. Beyond podosome biology, MT-FLIM and PC probes will be useful in studying immune cell interactions, Notch-Delta signaling, and adherens junctions[42,78,79]. Fluorescence lifetime is an improved indicator of density versus tension, and because hairpin probes unfold specifically under receptor tension, all measured forces can be attributed to integrins rather than to the vertical force balance vector that arises at the contact line between a liquid droplet applying pressure on a solid substrate[80]. In conclusion, we introduce and apply powerful DNA mechanotechnology tools to demonstrate the role and regulation of receptor forces on fluid substrates.

## Methods

**Probe synthesis and purification.** All oligonucleotides (Supplementary Table 1, Supplementary Fig. 1) were custom synthesized by Integrated DNA Technologies, except for BHQ1 modified oligos, which were custom synthesized by Biosearch Technologies. One hundred micrograms cyclo [Arg-Gly-Asp-D-Phe-Lys(PEG-PEG)] (PCI-3696-PI, Peptides International) was sonicated with 50 μg NHS-azide (88902, Thermo Fisher Scientific) in 10 μL dimethyl sulfoxide (MX1457-7, Millipore-Sigma) for 1 h. The azide-modified cyclo [Arg-Gly-Asp-D-Phe-Lys(PEG-PEG)] peptide was purified via reverse-phase high performance liquid chromatograph (HPLC) with a Grace Alltech C18 column (0.75 mL min⁻¹ flow rate; solvent A: nanopure water + 0.05% trifluoracetic acid (TFA), solvent B: acetonitrile

(BDH83639.400, VWR) + 0.05% TFA; starting condition 90% A + 10% B, 1% per min gradient B) (Supplementary Fig. 2a). Following HPLC purification, products were dried in an Eppendorf Vacufuge plus. Subsequently, the azide-modified peptide was ligated to DNA oligos containing a 5′ hexynyl modification using copper-catalyzed azide-alkyne cycloaddition. Briefly, 5 μL of 1 mM oligonucleotide was reacted for 1 h with ~30 nmol azido peptide in the presence of 10 mM sodium ascorbate, 1 mM copper sulfate, and 0.8 mM THPTA (1010, Click Chemistry Tools). The reaction was purged under $N_2$. The product was purified using reverse-phase HPLC with an Agilent Advanced oligo column (0.5 mL min$^{-1}$ flow rate; solvent A: 0.1 M TEAA, solvent B: acetonitrile; starting condition: 90% A + 10% B, 1% per min gradient B). Oligos were conjugated to Cy3B NHS ester (PA63100, GE Healthcare) in a 10 μL reaction; 50 μg excess Cy3B NHS was reacted with 2–5 nmol amine-modified DNA in 1× phosphate buffered saline (PBS) and 0.1 M sodium bicarbonate overnight. The product was purified using a P2 size-exclusion gel to remove excess dye prior to HPLC purification. Reverse-phase HPLC was performed with an Agilent Advanced oligo column as described above (Supplementary Fig. 2b–d). Final products were resuspended in nanopure water. If significant excess dye remained for MT-FLIM strands, the DNA was repurified with an amicon filter (Amicon Ultra-0.5 mL, Centrifugal Filters, Ultracel-3K) or with HPLC. Starting material and final masses were confirmed using matrix-assisted laser desorption/ionization-time of flight (MALDI-TOF) (Supplementary Fig. 3, Supplementary Table 2).

**Mass spectrometry.** Oligonucleotides in nanopure water (18.2 M Ω, Barnstead Nanopure) were plated in a 1:1 vol/vol ratio with saturated 3-hydroxypicolinic acid (56197, Millipore-Sigma) in 50% acetonitrile, 0.1% trifluoroacetic acid, and 5 mg mL$^{-1}$ ammonium citrate. Dried samples were massed with MALDI-TOF on a Bruker Daltonics ultraflex II TOF/TOF and analyzed using flexAnalysis 3.4.

**Small unilamellar vesicle (SUVs) preparation.** SUVs were prepared using a 10 mL LIPEX Extruder (Transferra Nanosciences, Inc.). Lipids were mixed in ~500 μL chloroform with 1,2-dioleoyl-sn-glycero-3-phosphocholine (DOPC) (850375C, Avanti Polar Lipids) as the base lipid. Biotinylated lipids were incorporated at 0.05–0.2 mol% 1,2-dioleoyl-sn-glycero-3-phosphoethanolamine-N-(biotinyl) (Biotinyl-Cap PE) (870282C, Avanti Polar Lipids). To directly tag the membrane in control experiments, N-(fluorescein-5-thiocarbamoyl)-1,2-dihexadecanoyl-sn-glycero-3-phosphoethanolamine, triethylammonium salt (FITC DHPE) (23304, AAT Bioquest) was incorporated at 0.1 mol%. Lipids were dried first by rotary evaporation and second by ultrahigh purity $N_2$. Lipid cakes were resuspended and sonicated in 3 mL nanopure water (final concentration, 2 mg mL$^{-1}$) prior to performing three freeze-thaw cycles. SUVs in nanopure water were then extruded 10× through a 0.08 μm polycarbonate filter (WHA110604, Whatman) supported by a drain disc (WHA230600, Whatman). SUVs were used within ~2 weeks.

**SLB Preparation.** Planar SLBs were prepared on either uncoated glass-bottom 96-well plates (265300, Nunc or 82050-782, Greiner) or glass coverslips (48382-085, VWR). Coverslips were washed and sonicated 3× in nanopure water followed by sonication in ethanol. Coverslips were dried at 90 °C overnight and cleaned in piranha solution (3:1 sulfuric acid and 30% hydrogen peroxide; caution, piranha acid is extremely corrosive and can explode if exposed to organic materials). Cleaned coverslips were washed 3× in nanopure water and were mounted into coverslip chambers in 1× PBS for SLB formation. Alternatively, 96-well plates were etched for 1–3 h in 2.6 M sodium hydroxide and were washed with 10 mL nanopure water and 5 mL 1× PBS. SLBs were formed by adding SUVs to etched glass for at least 5 min and were washed in nanopure water and 1× PBS prior to ~25 min blocking with 0.1% bovine serum albumin, Fraction V (10 735 078 001, Roche Diagnostics GmbH). Unless otherwise stated, all experiments were carried out with 99.9 mol% DOPC and 0.1 mol% Biotinyl-Cap PE. Blocked SLBs were washed with 5 mL 1× PBS and then saturated in 90–180 nM streptavidin (SA101, Millipore-Sigma) for at least 45 min. Excess streptavidin was removed with 10 mL 1× PBS, and SLBs were incubated with 30 nM DNA for at least 45 min. Functionalized SLBs were washed in 10 mL 1× PBS and then buffer exchanged into hanks balanced salts (Millipore-Sigma) for all imaging. To stain the membrane, SLBs were shaken for 30 min at 240 rpm with 10% (v/v) 250 μg μL$^{-1}$ b-BODIPY FL $C_5$-HPC (2-(4,4-difluoro-5,7, dimethyl-4-bora-3a,4a-diaza-s-indacene-3-pentanoyl)-1-hexadecaonyl-sn-glycero-3-phosphocholine (D3803, Thermo Fisher) or were incubated with 10% (v/v) 1.5 μg mL$^{-1}$ 1,1′-dioctadecyl-3,3,3′,3′-tetramethylindodicarbocyanine,4-chlorobenzenesulfonate salt (DiD) (D7757, Thermo Fisher) (DID). For MFM bead experiments, SLBs were assembled on 5 μm silica beads (SS06N, Bang Laboratories). One hundred microliters of 1 mg mL$^{-1}$ beads were rocked with 100 μL of DOPC SUVs. SLB-beads were washed in 1× PBS and purified 3× with centrifugation (5 min, 2000 rpm). Purified SLB-beads were incubated with 5 μM 1,1′-dioctadecyl-3,3,3′,3′-tetramethylindocarbocyanine perchlorate (DiI) (468495, Thermo Fisher) for 15 min. Free dye was removed by washing and pelleting 3× in 1× PBS (5 min, 2000 rpm)[49].

**DNA hybridization.** DNA oligonucleotides in 1× PBS were heated to 90 °C for 5 min and cooled at 25 °C for 25 min in a 0.2 mL thermowell tube. The ligand strand was added in 10% molar excess except for in absorbance spectroscopy and

PC experiments, in which strands were added in a ratio of 1:1:1. To chemically open tension probes, tension probes were hybridized with 10× molar excess complementary sequence (Supplementary Figs. 6 and 8a).

**UV-Vis spectroscopy.** Oligonucleotides (10 μL of 2.5 μM hairpin strand) were hybridized as described above to assemble closed tension probes in solution. Following hybridization, thermowell tubes were fit inside microcentrifuge tubes and dried in a vacufuge to <2.5 μL. The volume was adjusted to ~2.5 μL using nanopure water, corresponding to a final concentration of ~10 μM in 4× PBS. Three absorbance spectra per sample were collected with a pathlength of 0.1 cm on a Thermo Scientific Nanodrop 2000c spectrophotometer. This process was repeated in the presence of 10× molar excess of the complementary strand for each sample.

**DNA probe fluorescence calibration.** Closed and opened MT-FLIM probes were hybridized as described above and were combined in known ratios in solution immediately prior to incubation with SLBs. For 19 pN probe calibration, a scrambled hairpin was used in opened samples to lower the ΔG of hybridizaton[24]. SLB fluorescence intensity was measured using epifluorescence, and the background subtracted data were fit using linear regression to determine QE. Nineteen piconewton and 4.7 pN surfaces did not display statistically significant differences; thus, intensity data were combined for this calculation. For MTFM probes used in MFM and PCB experiments, QE was determined directly from background subtracted opened and closed images using the equation:

$$\text{Quenching Efficiency(QE)} = 100\% \times \frac{I_{\text{Open,BS}} - I_{\text{Closed,BS}}}{I_{\text{Open,BS}}}. \tag{1}$$

To determine the average fluorescence lifetime of MT-FLIM probes, surfaces containing 0–100% open probes were imaged using the same conditions as in cell experiments. For each image, an average fluorescence lifetime histogram was produced from defect-free regions of interest (ROI) in the image. Histograms were generated in SymPhoTime and contained the default 400 bins across a range of range of 0–12.5 ns. The corresponding ROI fluorescence decay had ~10$^5$ photons in the peak. Data were averaged across 2–3 surfaces per condition, and the resulting calibration curves of average fluorescence lifetime versus percent open probes were fit to a biexponential (Supplementary Note 1) and used to generate look-up tables.

**Probe density determination.** Relative probe density (clustering) per pixel was calculated according to the following equation (Supplementary Note 1):

$$\rho = \frac{I - D}{\frac{I_a - D}{1 - QE} F(\tau) + (I_o - D)(1 - F(\tau))} \tag{2}$$

Here $\rho$ is probe density, $I$ is the illumination profile corrected photon count, $D$ is dark counts, $I_o$ is the illumination profile corrected photon counts on the bilayer in the absence of the cell, QE is the quenching efficiency, and $F(\tau)$ is the percent of open probes. For SLBs with linear probes lacking a stem-loop structure, the percent open was assumed to be zero.

**Image processing.** Image processing was performed in MATLAB 2018a (MathWorks) using semiautomated custom scripts unless otherwise stated. Nikon ND2 files were directly transferred into MATLAB using the Bioformats Toolbox. Linescans were generated in MATLAB or ImageJ, and kymographs were generated in FIJI using the MultiKymograph plugin. The diffusion coefficient, $D$, was calculated according to the following equation[40]

$$D = w^2/4t_{1/2} \tag{3}$$

in which $w$ is the radius of the bleached region and $t_{1/2}$ is the time of 50% recovery determined by exponential curve fitting. MT-FLIM analysis methods are described in Supplementary Notes 1 and 2 and Supplementary Fig. 11. PC biotin time-lapse experiments were processed as shown in Supplementary Fig. 22. For LifeAct analysis, podosomes were identified by intensity-based thresholding of RGD depletion, and podosomes multiples were separated with a 2-pixel line. Clear single podosomes were selected by hand. For each cell, the data were normalized to the corresponding measurement from the podosome containing the brightest LifeAct signal. For time-lapse image analysis, podosome tracking was performed by hand, starting with the final frame and selecting the center of each podosome or corresponding cluster until its emergence or the time-lapse start. To track clusters, images were thresholded based on size and lack of local depletion zone in MATLAB and then tracked using TrackMate in FIJI. Cluster tracking did not allow merging or splitting events. For TGT analysis, because some 12 pN podosomes were challenging to identify computationally, podosomes were counted and identified manually. To determine percent depletion, the centroid of each podosome was selected and dilated by 2 pixels. Fluorescence intensity was then quantified within the dilated region.

MFM imaging was analyzed in MATLAB using published[49] code that was modified to threshold podosome rings. The XY orientation of the force was given by the phase of the sinusoidal fluorescence variation, while the amplitude of the sinusoidal variation gave information about the Z orientation. To determine the

anisotropy from emission-resolved polarization images, large numerical aperture correction factors were applied in MATLAB[81].

**Cell culture**. NIH 3T3 fibroblasts and mouse embryonic fibroblasts stably transformed with SrcY527F (MYF) were cultured in Dulbecco's Modification of Eagle's Medium (B003K32, Corning) with 10% fetal bovine serum (USDAFBS, MidSci), 2.5 mM L-glutamine (G8540, Millipore-Sigma), 1× penicillin streptomycin (97063-708, VWR). Cells were incubated at 37 °C in 5% $CO_2$. Cells were passaged at ~80% confluency. NIH 3T3 fibroblasts were gifted from colleagues at Emory University and were authenticated by IDEXX Bioresearch (Supplementary Fig. 24). Cells were imaged in Hanks Balanced Salts (H1387, Millipore-Sigma), pH 7.4. MYF cells were stably transfected and gifted by Olivier Destaing.

**Transfection**. NIH 3T3 fibroblasts were transfected in a 3:1 ratio of Lifeact-mCerulean-7 (#54713, Addgene) and polyethylenimine (23966-1, Polysciences, Inc.) (1 μg mL$^{-1}$, pH 7) in Optimem for 24 h prior to confocal imaging with the pinhole fully open. To knockdown *MYH9*, NIH 3T3 fibroblasts were transfected with Lipofectamine RNAiMAX (13778030, Thermo Fisher) and SMARTpool: ON-TARGETplus *MYH9* siRNA (D-040013-00-0005, Horizon Dharmacon) or ON-TARGETplus Non-targeting Pool (D-001810-10-05, Horizon Dharmacon) as a control. Transfection was performed according to the RNAiMAX manufacturers protocol in a six-well plate. Total volume was 2 mL, and media was changed ~48 h following transfection. Seventy-two hours following transfection, cells were assayed for podosome formation on SLBs using MT-FLIM and were plated on glass. Knockdown was functionally validated by assessing stress fiber formation in transfected cells ~12 h after plating.

**Immunostaining**. For staining on an SLB, NIH 3T3 fibroblasts were fixed and stained on 56 pN TGT probes following 2 h cell spreading. Cells were fixed in 2–4% formaldehyde in 1× PBS for 8–10 min. Cells were permeabilized for 3 min with 0.1% Triton X-100 and were blocked with BSA for 30 min. To characterize the relationship between FRET probe signal and integrin receptors in clusters and podosomes, cells were not permeabilized, and blocking was performed with a low concentration of BSA (0.01–0.1% w/v), as this best maintained the membrane and clusters. Staining was performed for 1 h at room temperature with 1:1000 Alexa 488-Phalloidin (ab176753, Abcam), 1:50 Vinculin Antibody SF9 647 (sc-73614 AF647, Santa Cruz Biotechnology), 1:50 Phospho-Paxillin (Tyr1888) Polyclonal Antibody (PA5-17828, Thermo Fisher) followed by 1:5500 Alexa Fluor 555 goat anti-rabbit (A21147, Thermo Fisher), or 10 μg/m Anti-Integrin β1 Antibody, clone MB1.2 (MAB1997, Sigma-Aldrich), followed by 1:1000 Alexa Fluor 647 goat anti-mouse IgG$_{2b}$ (γ2b) (Thermo Fisher) as indicated. Staining on an SLB requires gentle and gradual buffer exchange to avoid disruption of cells and DNA on the membrane. For staining of NIH 3T3 cells on glass, cells were not permeabilized and were stained with Alexa 488-Phalloidin for 30 min. Immunostained cells were imaged using total internal reflection fluorescence microscopy (TIRFM).

**Drug treatment**. Cells spread and formed podosomes on 4.7 pN MT-FLIM probes for ~1 h prior to imaging and addition of drug dissolved in DMSO (final DMSO concentration ~0.1%). Cells were either treated with 500 nM jasplakinolide (J7473, Thermo Fisher) for 8 min, with 50 μM Y27632 dihyrdochloride (Y0503, Millipore-Sigma) for 20 min, or with 50 μM (−)-blebbistatin (B0560, Millipore-Sigma) for 30 min. Blebbistatin was first heated to 45 °C in hanks balanced salts to improve solubility[82].

**Microscopy**. Epifluorescence and TIRFM were performed on a Nikon Eclipse Ti microscopes using the Nikon Elements 4.40.00 or 4.13.05 software and a 1.49 NA CFI Apo 100x objective. To maintain focus during time-lapse experiments, a Nikon perfect focus system was used. TIRFM images were collected with 80 mW 488, 561, and 647 nm lasers using a Chroma quad cube (ET-405/488/561/640 nm Laser Quad Band) or Chroma quad band C148022 and C-TIRF Cube C125986. RICM images were collected with Nikon cube 97270, and epifluorescence images were collected with Chroma cubes 49004 and C121664. For emission-resolved polarization imaging, the emission fluorescence was split into parallel and perpendicular channels using an Andor TuCam system with a wire grid polarizer (Moxtek, Andor: TR-EMFS-F03). Fluorescence was projected two Andor iXon Ultra 897 electron-multiplying charge-coupled devices. The polarization bias of the microscope (G-factor) was computed by taking the ratio of the parallel and perpendicular fluorescence emission of fluorescein in solution. To correct for the polarization scrambling effect of the objective, large numerical aperture corrections were applied to the raw fluorescence data as previously described[81]. To validate the ability of the microscope to measure systematic spatial variation in anisotropy produced by highly ordered fluorophores, the fluorescence anisotropy of 5 μm silica beads coated with DOPC SLBs doped with DiI was measured (Supplementary Fig. 16)[49].

Fluorescence lifetime and confocal measurements were performed on a Nikon Ti Eclipse Inverted confocal microscope with a Plan Apo Lambda 60×/1.40 Oil objective and Nikon Elements 4.40.00. Focus was maintained during timelapse imaging via a Nikon perfect focus system. The confocal microscope was equipped

with a C2 Laser launch with 405 and 561 nm diode lasers, Nikon Elements software, and a Picoquant Laser Scanning Microscope Time Correlated Single Photon Counting (TCSPC) Upgrade with SymPhoTime 64 2.1.3813. TCSPC settings are summarized in Supplementary Table 3. FLIM samples were excited with a 20 MHz pulsed 514 nm laser, and 512 × 512 pixel images were collected at 0.5 frames-per-second with 0.08 or 0.14 μm/pixel and an average photon count rate of ~4–6% of the laser pulse rate. Light was filtered using a long-pass laser beamsplitter (H560 LPXR, Analysentechnik) and a 582/75 bandpass filter (F37-582, Semrock). Laser light that was reflected by the dichroic was blocked from an additional detector using a 690/35 bandpass filter. Average fluorescence lifetime per pixel was calculated using the Fast FLIM algorithm in SymPhoTime. Only photons contributed by the long-pass detector (582/75) were considered in analysis. PC biotin experiments were performed on the confocal microscope as illustrated in Fig. 4c. Cells were imaged to identify a ROI for photocleavage. Then, time-lapse imaging was performed using the Nikon Elements Photostimulation Module. In phase 1, a 561 nm image was collected (−12 s). In phase 2, the ROI was photostimulated for five frames using a 405 nm laser (15%). In phase 3, the cell's response was tracked every 30 s by imaging in the 561 nm channel (0–180 s).

MFM[49] experiments were performed on a custom-built system (Supplementary Fig. 17a) consisting of a Nikon-T2 microscope with a 60 × 1.49 NA objective (Nikon), a cleanup plate, half-wave plate, and a focusing lens (ThorLabs), an Obis 561 nm LS 150 mW laser (Coherent), and an ORCA-Flash 4.0 v3 CMOS camera (Hamamatsu) The half-wave plate was rotated by a motorized mount (PRM1Z8; ThorLabs) driven by the Kinesis software. The half-wave plate was set in motion at 25° (rotating the excitation polarization by 50°). Once the half-wave place reached maximal rotational velocity, 73 images were taken at 50 ms exposure time via Nikon Fast acquisition in Nikon Elements v5.1. Each 50 ms exposure image contained the average fluorescence as the laser polarization rotates through 2.5° of arc.

**Podosome model**. To determine the net integrin tensile force exerted per podosome, $F_{Pod}$, we modeled a podosome with an outer radius, $R$, of 1 μm and an inner depletion radius, $r$, of 0.3 μm. Using the published value for the DOPC footprint within a membrane[83], we calculated the approximate number of biotinylated lipids per square micron, $l$. Based on our MT-FLIM data and % depletion data, we assumed that probes have a relative oligonucleotide density of 0.5–2 probes per biotinylated lipid and that integrin receptors exert $F_{Int}$ up to 50 pN. Relative probe density, $\rho$, in the podosome ring was assumed to be equal to that of the SLB background on average. To determine the magnitude of integrin forces in podosomes, we assumed 10% open probes per pixel, $O$, which was roughly equal to our measured data (Fig. 2). Per receptor integrin forces, $F_{Int}$, were modeled from 0 to 50 pN. Thus,

$$F_{Pod} = [\pi(R^2 - r^2)](ld)(\rho F_{Int}O). \quad (4)$$

We also plotted an experimental data point parameterized using the percent open data from Fig. 2. Because MT-FLIM probes are binary, they can provide the minimum applied force per receptor but not the absolute force per receptor. For this data point we set $F_{Int} = F_{1/2}$ with $F_{1/2}$ equals 19 pN (11% open) or 4.7 pN (3% open). The low force population was set by the difference in percent open probes on 4.7 and 19 pN MT-FLIM probes. Note that this data point represents the minimum force applied by the podosome on an SLB, because $F_{Int}$ is likely greater than $F_{1/2}$ and could also be a distribution. For modeling of a hypothetical podosome with isometric tension for emission-resolved polarization measurements, we modeled a 1 μm structure exerting inward tensile forces with a tilt angle of 20° (Supplementary Fig. 17c).

**Statistics and reproducibility**. Statistics were performed in MATLAB 2018a or in GraphPad Prism 7. For biological experiments, each experiment is defined as one flask or well of cells. For surface characterization, each experiment is defined as one SLB. Biological experiments were repeated at least three times, except for emission-resolved polarization imaging and immunostaining of β1 integrin with MT-FLIM probes and actin, which were repeated twice. Surface characterization was repeated at least twice per condition. For per-podosome data, only podosomes that were in clear focus, easily distinguishable, and that met thresholding criteria were analyzed. $P$ values are reported as ns $P > 0.05$, $*P < 0.05$, $**P < 0.01$, $***P < 0.0001$, $****P < 0.0001$. Detailed information on statistical tests, reproducibility, and outlier omission are listed in each figure caption.

**Reporting summary**. Further information on research design is available in the Nature Research Reporting Summary linked to this article.

## Data availability

Data supporting the findings of this manuscript are available from the corresponding authors upon reasonable request. A reporting summary for this article is available as a Supplementary Information file. The source data underlying Figs. 1b, d, e, 2b, c, e, f, 3d, 4b, d, 5f, 6d–g, 7a, b, and Supplementary Figs. 3a–c, 5c, f, i, 6c, d, 8, 10d, 13c, 15b, 18a, b, 19b, c, f–h, 20c, 21d, and 23c are provided as a Source Data file. The source data underlying Supplementary Fig. 10 are provided as a .mat file.

## Code availability

Semiautomated analysis codes are available on the Salaita Lab GitHub page: https://github.com/SalaitaLab.

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

## Acknowledgements

This work was supported by the NIH (R01-GM131099 and R01-GM124472) and the NSF CAREER Award (1350829). This material is based upon work supported by the National Science Foundation Graduate Research Fellowship Program under Grant No. DGE-1444932 (R.G. and J.M.B.) and by NCI-1F99CA234959-01 (J.M.B.). This study was supported in part by the Emory Comprehensive Glycomics Core, which is subsidized by the Emory University School of Medicine and is one of the Emory Integrated Core Facilities. Additional support was provided by the National Center for Advancing Translational Sciences of the National Institutes of Health under Award number UL1TR000454. This work was supported by the National Center for Advancing Translational Sciences of the National Institute of Health under Award number UL1TR002378. The content is solely the responsibility of the authors and does not necessarily reflect the official views of the National Institutes of Health. O.D. is funded by ANR agency and by LLNC as Equipe labellisée Ligue 2014 and by FRM as Equipe labellisée.

## Author contributions

R.G. and K.S. conceived of the project. R.G. designed and performed experiments and analysis. J.M.B. contributed to experiments, interpretation, and performed MFM analysis. E.B. and A.L.M. contributed to MFM experimental design, setup, and interpretation. O.D. and K.S. contributed to experimental design and interpretation. All authors contributed to preparation of the manuscript. R.G. and K.S. wrote the manuscript.

## Additional information

**Competing interests:** The authors declare no competing interests.

