## [Peer Review File · Nature Communications]

Reviewers' Comments:

Reviewer #1:

Remarks to the Author:

Glazier and colleagues investigate force generation and transmission in podosomes, actin- and integrin based adhesion and matrix degradation structures. As a model they use fibroblasts transformed with active Src, which is known to induce podosome formation. They further use a lipid bilayer system which supports podosome formation and forms the anchoring point for a variety of force-detecting probes that are employed during the course of the study.

Using various newly developed probes, the authors find that podosomes apply mostly vertical forces on the substrate. These are generated by actin polymerization of the core, which leads to integrin-based forces in the pN range on the podosome ring. The authors propose a model in which forces generated by actin polymerization at the core induce tension on the podosome ring structure, which is largely independent of myosin IIA activity, as commonly thought.

This is a very interesting paper that uses novel and clever technology to meticulously evaluate and measure force generation at podosomes. It challenges several current assumptions in the field and should thus spark new and fruitful discussions. Moreover, the developed probes should be highly interesting to the wider cell biological community. However, there are still several points that need to be addressed. Especially the limits and differences of the fibroblast model and lipid bilayer system used, in contrast to constitutively podosome forming cells and rigid substrates, should be clearly stated and discussed.

Major points:

- 1) The authors use fibroblasts transformed with active Src. This may not be the most appropriate model, as expression of constitutively active Src likely bypasses or alters some signal transduction pathways involved in proper podosome formation and turnover. This should at least be discussed. It is also not an "established model of podosomes...", as mentioned on p.4, as it combines features of both podosomes and invadopodia. So far, podosome-resembling structures on lipid bilayers have also only been addressed as "podosome-like". The authors should stick with this terminology.
- 2) P.7, Fig. 1d: Can the authors show that the structures termed "integrin clusters" really contain integrins, e.g. by using labeled constructs?
- 3) P.7, Fig. 2: The podosome ring is a clearly defined structure. Can the authors show that the ring-like force patterns they detect around podosomes, and which they address as "podosome rings" do actually colocalize with the adhesion plaque protein-containing ring? Localization of the forces relative to the ring structure seems to be important for their concept.
- 4) The authors propose that podosome-localized integrin forces are myosin IIA-independent. This is a central assumption in the field and should also be tested in myosin IIA-depleted cells.
- 5) Podosomes are known to show oscillatory protrusive behavior in the range of several seconds. Can this be detected in the set-up the authors use? If not, this should be discussed. Especially, as several groups (Cambi, Linder) have shown that this is a myosin IIA-based mechanism.
- 6) The cells used by the authors form only several podosomes and thus no widespread array of interlinked podosomes such as in myeloid cells. The latter, including macrophages, dendritic cells and osteoclasts, show a dense network of (myosin IIA-contractile) podosome-connecting actin cables. If the authors indeed want to conclude "...a minor contribution of mechanical coupling across the podosome network."), these cells would be appropriate models. Based only on using the Src-transformed fibroblasts, such a general conclusion can not be drawn.

Minor points:

- 1) "Proteolytic podosomes that mediate cancer invasion are termed invadopodia". This is not correct in several respects, as podosomes are by definition proteolytic, at least temporarily, and second, podosomes and invadopodia also differ in terms of architecture, lifetime and other aspects. Please rephrase.

- 2) P.2: (Fig. 1d) and (Fig. 1e,f) are actually (Fig. 2d) and (Fig. 2e,f)
3) Pp.17,18: (Fig. 8a,b) and (Fig. 8c) are actually (Fig. 6a,b) and (Fig. 6c)

Reviewer #2:

Remarks to the Author:

In this study entitled 'DNA mechanotechnology reveals that integrin receptors apply pN forces in podosomes on fluid substrates' the mechanics of podosomes are analyzed. Podosomes are integrin-mediated adhesion complexes that usually form in specialized cell types like osteoclasts and macrophages. The formation of podosomes was previously reported also in other cell types, for example fibroblasts, when cells were adhering to functionalized supported lipid bilayers (SLBs) upon which significant traction forces cannot be generated. Podosomes are characterized by a dense actin core surrounded with a ring-like complex comprising integrin receptors and their associated adhesion proteins. The actin core is thought to apply compressive force onto the underlying substrate, but it remained unclear to which degree contractile forces contribute to podosome mechanics, because suitable techniques to measure such forces were missing.

This manuscript reports the development of tools to visualize and quantify integrin forces in cells that adhere to SLBs and form podosomes. The authors use a combination of their previously published DNA sensor probes and fluorescence lifetime imaging (FLIM) to demonstrate tension on integrin receptors, polarization based molecular force microscopy to determine the orientation of mechanical forces, and the tension-gauge-tether approach to evaluate integrin force magnitudes during podosome formation. The data indicate integrin tension in the podosome ring, and forces appear to be oriented perpendicular to the substrate surface. Integrin forces in podosomes are largely insensitive to myosin activity but they depend on actin polymerization. The use of a photocleavable force probes indicates a local regulation of podosome mechanics by integrin receptors.

Overall, this is a very interesting study. The manuscript is written well and the work is technically appealing. However, there are some open questions regarding data analysis and data interpretation, and I feel that it would be important to address them before publication. Please see my specific comments below.

1. The authors apply FLIM microscopy to determine the percentage of opened DNA sensor probes. Owing to limited photon counts and multi-exponential fluorescence decays, the average lifetime is derived in a fit-free procedure from the detected photon arrival times. Based on the provided information, it is not clear how this particular lifetime value is obtained. I would be important to describe this procedure in much more detail. As the authors advertise this approach as a new technique (MT-FLIM), it should be also clear, which algorithms are used to determine the average lifetime values and to which degree this procedure allows quantitative force measurements.
2. To calibrate the measurement, different ratios of open vs. closed probes were recorded. Fig. 2b shows that lifetime values plateau when about 50% of the probes are opened and it seems that differences at 60-100% open probes cannot reliably be detected anymore. Since the error bars are hard to spot (maybe plot in black color), it should be indicated whether lifetime values corresponding to 60-100% of open probes are statistically different.
3. To investigate the generation of integrin forces at podosomes, the authors use time-lapse FLIM imaging and semi-automated data analysis. Using this protocol, the authors observe an increase in fluorescence lifetimes in the actin core indicating integrin tension (Fig. 2g and Suppl. Fig. 9c). This result is confusing, because the data suggest the presence of compressional forces (indicated by fluorescence depletion) and contractile forces (indicated by the lifetime increase) in the core region. As described in the supplement, this could be explained by the fact that podosomes are rather small and dynamic structures. Nevertheless, the results indicate that the spatio-temporal force maps are limited and this issue should be openly addressed in the main text.
4. A central conclusion of the paper is that integrins apply contractile forces in podosomes. As indicated by the authors, this finding contrasts previous studies. Since this appears to be a controversial issue, I wonder whether the here presented data allow a second line of

interpretation. Is it not possible that the massive compressional forces in the actin core squeeze down the SLBs (and lower the associated tension probe) thereby straining those DNA probes that are in the vicinity of the actin core? In this scenario, integrins would not apply forces on their ligands but rather be pulled upon by the deformed SLB. A few nanometers of SLB deformation could affect the energy transfer rates quite significantly. Given the controversial issue of integrin tension in podosomes, it seems crucial to experimentally rule out the possibility that the here observed lifetime changes are just a consequence of local SLB deformation.

Reviewer #3:

Remarks to the Author:

R. Glazier and colleagues report the development of a new type of DNA-based tension probes and evaluate the forces transmitted by fibroblasts adhering on supported lipid bilayers. In particular, they show that piconewton forces are transmitted by integrins surrounding podosome actin cores. The manuscript is well-illustrated and constitutes an important contribution to understand how forces are generated and transmitted by podosomes. However, some information should be added to strengthen the impact of the manuscript.

Main comments:

-The novelty of the DNA-based force probes is not sufficiently documented to support that it is really novel. I propose to add a paragraph in the introduction telling the state of the art on this aspect including previously published methods. Moreover, the characteristics, advantages and limits of the methods described in the present manuscript should be detailed in the discussion section.

-The authors state in the introduction and discussion that they “demonstrate that podosome-podosome mechanical crosstalk is minimal”. They show that blebbistatin has no effect on integrin forces and conclude from this result that podosomes are not mechanically connected. However, supported lipid bilayers are laterally fluid and impair transmission of lateral forces. To clearly demonstrate that there is no crosstalk between podosomes in fibroblasts, experiments +/- blebbistatin should be performed, either on nano-partitioned rigid lipid bilayers (as described in Yu CH, Cell Reports 2013), or on rigid substrates.

-At the end of the introduction, the authors state that they “provide direct evidence that podosomes are mechanosensors with local pN sensitivity”. Podosome mechanosensitivity has been evaluated in other articles, but not in this manuscript. To conclude that, tension measurements should be performed with fibroblasts layered on substrates of controlled stiffness.

-Results/ Podosome actin content correlates with RGD-probe depletion on SLBs:

Fig1: “The depletion of RGD-DNA was caused by pushing forces generated by the protrusive podosome core”.

This interpretation is not supported by the experiments. To ascertain that depletion is due to protrusion, the axial deformation of the SLB should be shown and quantified.

-Results / MT-FLIM maps pN receptor tension and clustering on SLBs:

“fluorescence lifetime is sensitive to energy transfer but independent of the dye concentration⁵⁰”.

Can the authors rule out that there is no FRET between two probes as a result of a higher concentration of the probes around podosome cores ? How does the calibration curve vary with probe concentration ?

-Results / MT-FLIM maps pN receptor tension and clustering on SLBs:

“Within 20 min of plating, cells formed dynamic integrin clusters which lacked detectable forces.”

These results are not shown. How were integrin clusters observed ?

-Results/ Actin protrusion and integrin tension are engaged in a mechanical feedback loop:
"we engineered "releasable" MTFM probes by anchoring probes with a photocleavable (PC) biotin group"

As a control of the effect of the 405 nm laser, tension in podosomes should be measured in the absence of photocleavable biotin groups.

Minor comments:

-The authors state that "pharmacological inhibitors show that forces are the direct consequence of actin-core polymerization". However, the drugs used do not specifically target actin-core polymerization, but more generally the polymerization of actin in the whole cell. I suggest to reformulate this conclusion.

-All along the manuscript, the authors use the term "contraction force" when mentioning the force applied at the podosome ring. "Contraction" implies that there is a (myosin II) contractility-based mechanism, which is in contradiction with the results of this manuscript since blebbistatin has no effect. I suggest to replace "contraction force" by "traction force".

Introduction section:

-The term "organelles" should be replaced, e.g. by "structures", as podosomes are not surrounded by lipid bilayers.

Result section:

-Results/ Podosome forces are primarily vertical:

Fig3c: It would be interesting to evaluate how the angle varies with the distance to the actin core ?

Discussion section:

-Fig.6 is numbered Fig.8 in the text.

-"the nature of the mechanical linkage between the substrate and the cytoskeleton was unknown^{16,19,27,30,60}"

In ref. 27, talin and vinculin, two proteins making mechanical links between integrins and actin have been identified.

Reviewer 1:

Comment 1: The authors use fibroblasts transformed with active Src. This may not be the most appropriate model, as expression of constitutively active Src likely bypasses or alters some signal transduction pathways involved in proper podosome formation and turnover. This should at least be discussed. It is also not an “established model of podosomes...”, as mentioned on p.4, as it combines features of both podosomes and invadopodia. So far, podosome-resembling structures on lipid bilayers have also only been addressed as “podosome-like”. The authors should stick with this terminology.

Response: Thank you for this comment. We agree that this is an important distinction that should be clarified in our work. As such, we have reworded our manuscript to follow the literature precedent set forth by Sheetz and coworkers (Yu, et. al. *Cell Rep.* 2013). We have amended the main text of the manuscript to clarify that these structures are “podosome-like” (see below). Additionally, we would like to apologize for any confusion in the paper. All the data presented in the main figures is from NIH 3T3 cells that are not Src-transformed. Only one single supplementary information figure (**Supplementary Fig. 12**) shows data using fibroblasts that had constitutively active Src. The purpose of this additional data was to demonstrate that invadosomes are amenable to imaging by MT-FLIM and that these structures produces similar spatial patterns of tension.

Revised text (p. 4): *“These adhesions were actin rich, excluded the RGD-DNA ligand (Fig. 1c) and were surrounded by vinculin and phospho-paxillin (Supplementary Fig. 4). Since these podosome-like adhesions contained the key elements of podosomes¹, we refer to these structures as podosomes throughout this manuscript.”*

Comment 2: Fig. 1d: Can the authors show that the structures termed “integrin clusters” really contain integrins, e.g. by using labeled constructs?

Response: To address this concern, we immunostained $\beta 1$ integrins to identify the location of integrins in cells. The data is presented in a new supplementary figure (**Supplementary Fig. 9**, included below) and shows that actin is anti-localized with $\beta 1$ integrins. Additionally, regions of tension and RGD-ligand clustering are localized with the $\beta 1$ signal. Based on literature precedent, it has been shown that $\beta 1$ integrins are a key regulator of invadosome function (Destaing, et. al. *Mol. Biol. Cell.*, 2010) thus justifying our choice for staining.

Revised Text (p. 7): *“Both clusters and podosome rings colocalized with $\beta 1$ integrin, confirming that they were caused through integrin-mediated adhesions (Supplementary Fig. 9.”*

New Supplementary Figure:

Comment 3: Fig. 2: The podosome ring is a clearly defined structure. Can the authors show that the ring-like force patterns they detect around podosomes, and which they address as “podosome rings” do actually colocalize with the adhesion plaque protein-containing ring? Localization of the forces relative to the ring structure seems to be important for their concept.

Response: This is a good point, which we have addressed through two experiments and supplementary figures. First, we performed staining (shown above) with our tension probes and anti-β1 integrin. This data shows colocalization of integrin receptors and the probe signal. Because this is an intensity-based experiment, note that the FRET probe signal reports the convolution of both tension and clustering. These data are now included in **Supplementary Fig. 9**, shown above. In addition, we bolstered our staining data to better confirm the adhesion proteins in the ring (**Supplementary Fig. 4**). Previously we presented staining for actin (core), vinculin (ring), and RGD (ring, with DNA probes). We now also include a panel with multiplexed staining for phospho-paxillin (pY118), vinculin, and β1 integrin confirming that the podosome ring complex includes adhesion-associated proteins (see below). This is consistent with the literature and with our proposed model of mechanotransduction.

New Supplementary Figure Panel:

Comment 4: The authors propose that podosome-localized integrin forces are myosin IIA-independent. This is a central assumption in the field and should also be tested in myosin IIA-depleted cells.

Response. To further strengthen the results obtained using blebbistatin treatment of cells, we now include siRNA experiments to knockdown *MYH9*, which is the heavy chain of myosin IIA. Knockdown was validated by imaging stress fiber formation of transfected cells on glass, as MIIa is required for stress fibers. We did not observe any significant changes in podosome size, and tension when cells were treated with a nontargeting siRNA versus si *Myh9*. These results agree with the literature, in which knocking down *Myh9* did not affect podosome formation (Rafiq, et. al. *J. Cell Biol.*, 2017). We now include a new supplementary figure (**Supplementary Fig. 19**) summarizing these results along with our blebbistatin data. The new panels pertaining to *Myh9* knockdown are shown below.

New text (p. 10) : “*We further validated this result by knocking down MYH9, the myosin IIA head domain. No significant changes in podosome depletion or tension were observed when MYH9 was knocked down (Supplementary Fig. 19).*”

Comment 5: Podosomes are known to show oscillatory protrusive behavior in the range of several seconds. Can this be detected in the set-up the authors use? If not, this should be discussed. Especially, as several groups (Cambí, Linder) have shown that this is a myosin IIA-based mechanism.

Response: This is a good point, which has also captured our interest. Podosome oscillations have only been shown on rigid substrates^{2,3} and have not been reported on the SLB model. Thus, the existence of such oscillations is an open question and one that is interesting to pursue. Unfortunately, the current MT-FLIM time-resolution prevents the observation of tension oscillations at the tens of seconds time scale. **Supplementary Fig. 10** shows the effect of photon count on signal/noise and suggests that we are unable to detect these oscillations. We hope to address this issue in future work where we improve the temporal resolution of MT-FLIM beyond what is possible with our current laser and electronic set-up.

Comment 6: The cells used by the authors form only several podosomes and thus no widespread array of interlinked podosomes such as in myeloid cells. The latter, including macrophages, dendritic cells and osteoclasts, show a dense network of (myosin IIA-contractile) podosome-connecting actin cables. If the authors indeed want to conclude "...a minor contribution of mechanical coupling across the podosome network."), these cells would be appropriate models. Based only on using the Src-transformed fibroblasts, such a general conclusion cannot be drawn.

Response: Thank you for this comment, which has highlighted a need for clarification within our paper. We agree that the data is specific for the 3T3 cell line model tested here. The myeloid cell lines with podosome-connecting actin cables may show different mechanical phenotypes. Therefore, we have amended the main text and our conclusions to emphasize this point.

Revised Text (p. 11): "These results confirm local mechanical feedback between integrin tension and actin protrusion as the primary mechanism of force *balance in podosomes on*

SLBs in this 3T3 cell line and suggest that there may be a minor contribution of mechanical coupling across the podosome network. Other podosome-forming cell types and especially myeloid cells may display different levels of podosome-podosome coupling.”

Comment 7: “Proteolytic podosomes that mediate cancer invasion are termed invadopodia” This is not correct in several respects, as podosomes are by definition proteolytic, at least temporarily, and second, podosomes and invadopodia also differ in terms of architecture, lifetime and other aspects. Please rephrase.

Response: In shortening this sentence we sacrificed scientific accuracy – thank you for bringing this to our attention. We realize that podosomes and invadosomes have many unique intricacies that prevent their uniform classification. To address this, we have reworded this sentence in our introduction.

Revised text (p. 2): *“Invadopodia, which are structurally similar to podosomes, facilitate cancer cell migration during metastasis.”*

Minor issues: 2) P.2: (Fig. 1d) and (Fig. 1e,f) are actually (Fig. 2d) and (Fig. 2e,f) 3) Pp.17,18: (Fig. 8a,b) and (Fig. 8c) are actually (Fig. 6a,b) and (Fig. 6c).

Response: These have been corrected in the updated manuscript.

Reviewer 2:

Comment 1: The authors apply FLIM microscopy to determine the percentage of opened DNA sensor probes. Owing to limited photon counts and multi-exponential fluorescence decays, the average lifetime is derived in a fit-free procedure from the detected photon arrival times. Based on the provided information, it is not clear how this particular lifetime value is obtained. I would be important to describe this procedure in much more detail. As the authors advertise this approach as a new technique (MT-FLIM), it should be also clear, which algorithms are used to determine the average lifetime values and to which degree this procedure allows quantitative force measurements.

Response: Thank you for this comment. The average fluorescence lifetime is generated during acquisition in Picoquant SymPhoTime 64 and represents center-of-mass lifetime for photons per pixel. In an effort to better describe our methods, we have added new details about our acquisition, analysis, and interpretation to our methods and to **Supplementary Note 1** on FLIM data analysis (shown below). We have also added a Supplementary Table containing TCSPC parameters.

Revised text (p. 19): “For each image, an average fluorescence lifetime *histogram* was produced from defect-free regions of interest (ROI) in the image. *Histograms were generated in SymPhoTime and contained the default 400 bins across a range of range of 0 to 12.5 ns. The corresponding ROI fluorescence decay had ~10⁵ photons in the peak. Data were averaged across 2-3 surfaces per condition, the and resulting calibration curves of average fluorescence lifetime versus percent open probes were fit to a biexponential (Supplementary Note 1) and used to generate look-up tables.*”

Revised text (p. 23): “Fluorescence lifetime and confocal measurements were performed on a Nikon Ti Eclipse Inverted confocal microscope with a Plan Apo Lambda 60X/1.40 Oil objective and Nikon Elements 4.40.00. Focus was maintained during timelapse imaging via a Nikon perfect focus system. The confocal microscope was equipped with a C2 Laser launch with 405nm and 561nm diode lasers, Nikon Elements software, and a Picoquant Laser Scanning Microscope Time Correlated Single Photon Counting (TCSPC) Upgrade with SymPhoTime 64 2.1.3813. TCSPC settings are summarized in **Supplementary Table 3**. FLIM samples were excited with a 20 MHz pulsed 514 nm laser, and 512 x 512 pixel images were collected at 0.5 frames-per-second with 0.08 or 0.14 μm/pixel and a photon count rate of ~4-6% of the laser pulse rate. Light was filtered using a long-pass laser beamsplitter (H560 LPXR, Analysentechnik) and a 582/85 bandpass filter (F37-582, Semrock). Laser light that was reflected by the dichroic was blocked from an additional detector using a 690/35 bandpass filter. Average fluorescence lifetime per pixel was calculated using the Fast FLIM algorithm in SymPhoTime. Only photons contributed by the long-pass detector (582/35) were considered in analysis.”

New Supplementary Table:

General	
TCSPC Resolution	25.0 ps
TCSPC Mode	T3
Sync	
CFD Level	-150 mV
Zero Cross	-10 mV
Sync Divider	8

Detector	
CFD Level	-45 mV
Zero Cross	-10 mV
Offset	530 ps

Revised Supplementary Note 1: Determination and interpretation of percentage open probes and local probe density

The intensity of tension probe signal, I , measured as the number of photon counts per pixel, is a function of the local probe density, ρ , the fraction of open and closed probes, O and C , their per-probe photon count contributions, m and n , and the dark counts,

$$I(O, C, \rho) = \rho(m(O) + n(C)) + D \quad (1)$$

Because probe opening is binary (probes are either open or closed), O and C are related such that:

$$O + C = 1 \quad (2)$$

Assuming that in the SLB background, all probes are closed and the relative probe density is 1, we find that:

$$I(0, 1, 1) = n + D = I_0 \quad (3)$$

Although m and n cannot be measured directly, they are related by the quenching efficiency, QE such that:

$$QE = \frac{m-n}{m} \quad (4)$$

$$m = \frac{n}{1-QE} \quad (5)$$

Rearranging (1) and substituting with (2)-(5), we find that probe density,

$$\rho = \frac{I(O, C, \rho) - D}{\left(\frac{I_0 - D}{1 - QE}\right)(O) + (I_0 - D)(1 - O)} \quad (6)$$

We obtained QE from the slope of the linear curve-fit epifluorescence images of titrated open and closed probes (**Figure 2c**). To determine D , we calculated the average number of photons per pixel on the identical second detector blocked with a 690 nm band-pass filter during calibration image acquisition. Since the average number of photons was less than 1 per pixel, we determined dark counts to be negligible. To determine O , the fraction of open probes per pixel, we used a 5%-open-interval look-up-table (LUT) generated from the empirical calibration curve of percent open probes versus average fluorescence lifetime per pixel, τ , (**Figure 2b**, **Supplementary Figure 7d,e**), which fit to the biexponential equation:

$$\tau = ae^{bO} - ce^{dO} \quad (7)$$

Here τ is the mean photon arrival time determined by the Fast FLIM algorithm in SymPhoTime 64. This metric provides the distance between the rise of the IRF and the center-of-mass of photon arrivals in a decay and does not require curve-fitting. This method has some disadvantages such as uncertainty due to uncertainty of time-zero and contribution of background photons; therefore, it is not typically the most quantitative metric to characterize a

system and is typically used to give an initial real-time FLIM image. However, it has still proven useful as an estimate of the fluorescence lifetime in a variety of FLIM applications⁴⁻⁶. For our purposes, we found it to be a robust measure of the fluorescence lifetime of our tension-probe surfaces, in which empirically measured lifetimes were much more important than the physical processes giving rise to these exact lifetimes. When used with our calibration curve to convert the average fluorescence lifetime per pixel to the percentage of open probes per pixel, the precision is related to the number of photons in the calibration curve image, which we held at constant of approximately 10^5 photons in the peak of the fluorescence lifetime decay curve, as well as with the number of photon counts in the pixel of interest. The major sources of uncertainty in this metric comes from the width of the histogram used to generate the percentage of open probes look-up-table and uncertainty due to the computed IRF. Because we were concerned that dwell time could cause some probe melting and affect the fluorescence lifetime, we performed these measurements for all imaging conditions (0.14 and 0.08), but this effect was negligible. The constants $a - d$ were as follows:

Probe (pN)	a	b	c	d	r ²
4.7	2.306	0.001	-1.166	-0.072	0.9996
19	2.295	0.002	-1.21	-0.056	0.9992

While we found the average fluorescence lifetime of probes on our SLBs to be consistent, it was important to minimize the free dye in solution, which could shift the lifetimes. We recommend that anyone using this method generate a calibration curve on their own instrument prior to data quantification.

Comment 2. To calibrate the measurement, different ratios of open vs. closed probes were recorded. Fig. 2b shows that lifetime values plateau when about 50% of the probes are opened and it seems that differences at 60-100% open probes cannot reliably be detected anymore. Since the error bars are hard to spot (maybe plot in black color), it should be indicated whether lifetime values corresponding to 60-100% of open probes are statistically different.

Response: Thank you for this comment. In the updated manuscript, we have now reformatted our graph to better display the error bars. For consistency, these changes have also been applied to **Fig. 2c**. We also clearly explain the meaning of the error bars associated with the average lifetime measured for each data point here in the caption. The error bars represent the standard error of the mean obtained with two to three different surfaces where each surface was measured three times and histograms were fit to a Gaussian distribution. Because the mean lifetime localizes precisely, error bars were small and hence were difficult to display in the plot. To indicate whether the average lifetime values assigned for the calibration curve are statistically different, we ran an ANOVA across the data set followed by a multicomparison test. To avoid graph congestion, we show the significance for data points with $p > 0.01$. All other interactions had $p < 0.01$, with most data points having $p < 0.0001$. Error bars have accordingly also been added to the corresponding SI plot for calibration with different zoom settings. Exact p values are included in the source data file. As suggested by the reviewer, our ability to distinguish pixels that have open probe densities from 60 to 100% is not strong. That said, we find that the vast majority of podosome-containing pixels show a mean % open value of 10-30%, so this is not the main source of uncertainty when determining the percent open probe density.

Revised Figure Panels:

Comment 3. To investigate the generation of integrin forces at podosomes, the authors use time-lapse FLIM imaging and semi-automated data analysis. Using this protocol, the authors observe an increase in fluorescence lifetimes in the actin core indicating integrin tension (Fig. 2g and Suppl. Fig. 9c). This result is confusing, because the data suggest the presence of compressional forces (indicated by fluorescence depletion) and contractile forces (indicated by the lifetime increase) in the core region. As described in the supplement, this could be explained by the fact that podosomes are rather small and dynamic structures. Nevertheless, the results indicate that the spatio-temporal force maps are limited and this issue should be openly addressed in the main text.

Response: We agree that increase in probe lifetime within the podosome core is confusing. As mentioned, in our supplementary note, the core of the podosomes is depleted of probes, and this leads to low photon count pixels, which through a combination of factors, exhibit high fluorescence lifetime. We suspect that this is the combination of detector noise and the effects of the diffraction limit and dynamics. Podosomes are spatially and temporally dynamic on the SLB substrate. Further compounding this issue, podosomes are often diffraction limited in size and hence the small photon count in the core may be due to contamination from the ring complex. Thus, these pixels may appear to have long lifetimes. A few photons with long lifetimes could skew the average fluorescence lifetime of a pixel that is particular dark. We now emphasize this limitation of our method in the main text and clarify our interpretation in **Supplementary Note 2**.

Revised Text (p. 7): *Note that low photon count pixels, such as within depletion zone, produce less reliable lifetime values; these pixels were excluded according to lifetime and photon count cutoffs described in **Supplementary Note 2 (Supplementary Fig. 11)**. Images represent the accumulation of signal over one minute, so there is some spatiotemporal averaging. Nevertheless, these images provide the first quantitative maps of integrin forces on an SLB and within podosomes.*

Revised Supplementary Note 2: *“We attributed this effect to three factors: (1) Podosomes are micron-sized structures and are subject to the diffraction limit. Thus, high fluorescence lifetime photons from the ring could be collected in depletion regions.*

(2) Podosomes are dynamic structures. Any movement of the ring could cause slight blurring of signal. (3) Depletion regions have low signal-to-noise ratio (SNR) and are subject to influence by noise (detector shot noise and otherwise). We hypothesize that these three features together give rise to the appearance of tension in the podosome core. Considering a primarily depleted (dark) podosome core, some photon contribution from the ring will increase the photon count, but it will still be much darker than the surrounding regions. However, assuming that most of these photons have a long fluorescence lifetime, then it will appear that the core region has a high percentage of open probes. This effect would be magnified by any movement or changes of the podosome ring or core structure during imaging.

To determine which pixels were the most reliable, we performed an extensive analysis of MT-FLIM photon statistics.”

Comment 4. A central conclusion of the paper is that integrins apply contractile forces in podosomes. As indicated by the authors, this finding contrasts previous studies. Since this appears to be a controversial issue, I wonder whether the here F data allow a second line of interpretation. Is it not possible that the massive compressional forces in the actin core squeeze down the SLBs (and lower the associated tension probe) thereby straining those DNA probes that are in the vicinity of the actin core? In this scenario, integrins would not apply forces on their ligands but rather be pulled upon by the deformed SLB. A few nanometers of SLB deformation could affect the energy transfer rates quite significantly. Given the controversial issue of integrin tension in podosomes, it seems crucial to experimentally rule out the possibility that the here observed lifetime changes are just a consequence of local SLB deformation.

Response 4: The SLB consists of a ~5 nm membrane that is supported on glass, with a ~1 nm hydration layer separating the membrane from the solid support⁷. According to the literature, the compression and bending of a supported bilayer are intrinsically linked, but the bilayer is resistant to compression and can be considered an incompressible fluid^{8,9}. Thus, the bilayer tends to fail rather than to become compressed. The literature as well as our own data strongly argues against SLB rupture in our work. Specifically, AFM studies show that the forces required to disrupt the SLB requires ~MPa pressures (~nN range forces focused on a contact area of a few hundred nm²). This level of pressure far exceeds those generated by the podosome core:

Parameter	Value	Method	Source	Calculated Pressure
SLB puncture force	58 nN	Biomimetic Stealth Probe (radius = 100 nm)	¹⁰	~1.8 MPa
Podosome protrusion	~10-50 nN	Model parameterized from tension probe data (radius = 300 nm)	This manuscript, Fig. 6	~0.4 – 0.18 MPa

Our experiments show that the SLB remains fluid and intact following podosome formation. This evidence comes from new membrane staining experiments and further quantification of our original data, as well as careful examination of our timelapse videos of podosome dynamics (**Supplementary Movies 1-3**). Moreover, membrane staining experiments confirm that the membrane is not punctured and disrupted following podosome formation (**Supplementary Fig. 5**).

Nevertheless, we have also performed an experiment to attempt to show that forces are transmitted through integrin receptors. In this experiment, we allowed cells to form podosomes on bilayers that were either decorated with a combination of:

1) tension probes containing ligands but that were spectroscopically silent (no fluorophore) and FRET probes lacking a ligand that could fluoresce but not bind to receptors or

2) tension probes that contained FRET labels and ligands and probes that were spectroscopically silent and did not contain a ligand (positive control group).

Podosomes formed and probes were excluded from the podosome core in both cases, but we only observed a change in fluorescence lifetime due to unfolding in the positive control, in which integrin receptors bound to cells.

Revised Text (p. 5): *“The depletion of RGD-DNA was caused by pushing forces generated by the protrusive podosome core¹. Note that these forces exclude RGD-DNA but do not rupture the underlying SLB (Supplementary Fig. 5), which is largely incompressible^{8,9,11}.”*

Revised Text (p. 8): *“In addition, control experiments in which the fluorophore-quencher pair and ligand were separated onto two-different co-presented DNA probes showed no change in fluorescence lifetime (Supplementary Fig. 13). Together, these data confirm that changes in lifetime are due to integrin-mediated mechanical unfolding of the hairpin.”*

New Supplementary Figures:

Reviewer 3:

Comment 1: The novelty of the DNA-based force probes is not sufficiently documented to support that it is really novel. I propose to add a paragraph in the introduction telling the state of the art on this aspect including previously published methods. Moreover, the characteristics, advantages and limits of the methods described in the present manuscript should be detailed in the discussion section.

Response: Thank you for this suggestion. We have added revised our introduction and our discussion about state-of-the-art methods in podosome mechanobiology and the new features that our methods offer. We also added a supplementary figure, which shows probe absorbance spectra, highlighting the novelty of this specific DNA-based force probe, which is uniquely suitable for time-resolved fluorescence. We found that conventional tension probes were static quenched and unsuitable for FLIM-FRET. Accompanying this update, we have changed the language from “conventional” to “static quenched” to describe these probes throughout our text.

Revised Introduction (p.2): *“Podosomes are specialized acto-adhesive structures that coordinate a variety of cell-type specific processes ranging from forming the sealing-zone for bone-resorption in osteoclasts to facilitating migration and antigen scavenging in immune cells¹²⁻¹⁵. In Wiscott-Aldrich Syndrome, cells fail to form podosomes, and patients experience frequent infections, impaired blood-clotting, and altered bone resorption¹⁶⁻¹⁹. In HIV, however, numerous enlarged podosomes are associated with increased cell migration, macrophage tissue infiltration, and elevated bone degradation^{20,21}. Invadopodia, which are structurally similar to podosomes, facilitate cancer cell migration during metastasis²²⁻²⁴. Hence, podosome formation and regulation is critical to disease pathophysiology and homeostasis.*

Akin to the widely studied focal adhesions (FA), podosomes have been shown to exert mechanical forces and to respond to ECM stiffness²⁵⁻³¹. Whereas FAs assemble into fibrillar micro-scale structures that apply contractile forces to the substrate^{25,32-36}, podosomes assemble into a columnar architecture consisting of an actin core surrounded by a ring complex containing adhesome proteins including integrin receptors³⁷. The actin core protrudes from the cell body, applying nN compressive forces onto the underlying substrate^{3,27,38}. Given that a static cell cannot experience a force imbalance³⁹, it is widely recognized that podosomes apply opposing tensile forces, with some disagreement as to the requirement for integrin adhesion forces^{3,37,38,40-43}. Mathematical modeling suggests that these tensile forces are localized to the podosome ring³⁸, and there are two lines of experiments that support this model. The first comes from biophysical measurements of talin extension²⁹ and vinculin tension³ within podosomes. These measurements are indirect as they fail to map the molecular forces applied by the podosome itself. The second class of measurements reports bulk substrate deformations using traction force microscopy (TFM). While TFM provides a direct measure of cell stresses, the spatial and force resolution of the method precludes mapping the forces at the podosome ring complex. A more sensitive variant of TFM that is interferometry-based offers improved force sensitivity but still averages deformations of the substrate⁴⁴, and thus cannot quantify receptor forces. To the best of our knowledge, no quantitative force maps have been reported validating the role of adhesion receptor mechanics in opposing actin protrusion and mechanically linking the substrate and the cytoskeleton within podosomes.

Further confounding podosome mechanical models, recent results demonstrated the formation of podosome-like adhesions on supported lipid bilayers (SLBs)^{1,45}. SLBs are phospholipid membranes that self-assemble onto a glass slide. Lipids are confined in the z-direction but are laterally fluid⁸. Thus, on SLBs, podosomes are reported to form even

in the absence of traction forces^{1,46}, which is confounding since podosomes apply compressive forces on the SLB.

In this work, we employ novel DNA-based mechanotechnology tools to challenge the hypothesis that integrins cannot apply forces on fluid membranes and to investigate the role of integrin tensile forces in podosome mechanosensing. As a material, DNA offers the ability to map and perturb receptor forces with pN resolution and sub-micron spatial resolution. We first quantify podosome-mediated depletion as a marker of actin core protrusion on SLBs. Next, we introduce Molecular Tension – Fluorescence Lifetime Imaging Microscopy (MT-FLIM), producing pN maps of integrin tension during receptor-ligand clustering at the cell membrane. Previously, molecular tension fluorescence microscopy (MTFM) imaging on supported bilayers was carried out using ratiometric probes^{47,48}, but these induced artificial clustering or employed three-way energy transfer, which hinder quantitative analysis. Moreover, we found that conventional MTFM probes³⁴ are static quenched and unsuitable for FLIM. Thus, we report a re-engineered DNA-based FLIM-FRET probe that circumvents these problems. To better understand podosome tension forces, we apply a newly developed force-orientation analysis technique to demonstrate that integrin forces are primarily perpendicular to the substrate. Treatment with pharmacological inhibitors showed that podosome tensile forces are a direct consequence of actin polymerization.

The latter section of our paper focuses on employing DNA mechanotechnology tools to manipulate podosome forces and reveal mechanoregulation. Specifically, using photocleavable DNA, we introduced local mechanical defects within podosomes with sub-cellular resolution, allowing us to track changes in tension and protrusion. These experiments demonstrate that podosome-podosome mechanical crosstalk is minimal within the cell models studied and that force balance is controlled locally. With tension-gauge tethers (TGTs), which limit the maximum magnitude of integrin tension, we demonstrate that podosomes not only apply integrin forces in their rings, but that these forces are required to stabilize podosomes.

Finally, we model pN integrin forces in podosomes and demonstrate that podosomes exert nN vertical forces on an SLB, in agreement with previously measured protrusion forces²⁷. Taken together, our work offers the first receptor-level quantitative maps of integrin tension on fluid substrates and provides direct experimental evidence that podosomes are mechanosensors with local pN sensitivity.”

New Results (p. 6): “The bottom arm was hybridized to a biotinylated quencher strand, containing an internal deoxythymidine BHQ1 modification. We selected this site for the quencher, to ensure the probe that the probe was FRET quenched, as absorbance spectroscopy demonstrated that conventional MTFM probes are static quenched and thus unsuitable for FLIM (Supplementary Fig. 6).”

Revised Discussion (p. 14): “Our results should be considered in the context of a few important caveats. While fibroblasts provide a robust model to study podosomes in vitro, to our knowledge, fibroblasts have not been shown to form podosomes in vivo. Another corollary point is that the mechanical properties and the stiffness of the substrate will likely influence podosome dynamics. Finally, while MT-FLIM quantitatively maps integrin tension on fluid substrates, this signal is subject to some spatiotemporal convolution, and fast changes in tension will be obscured during the one-minute acquisition. Furthermore, regions of low photon counts such as podosome depletion regions may be disproportionately affected by the diffraction limit. These issues will be addressed as superresolved and faster FLIM electronics become more widely available.

Nevertheless, our work provides valuable mechanobiology tools and insight. SLBs offer a unique landscape to study the minimal mechanical machinery required for podosome formation, and DNA probes provide a novel method to map and manipulate molecular forces. While RGD ligands on an SLB are more mobile than in physiological ECM, their mobility can recapitulate degraded ECM⁴⁹, which is relevant to podosome and invadosome biology^{12,24}. During cancer cell invasion, cells alternate between periods of integrin and invadosome-mediated matrix degradation and migration^{40,50}. Thus, podosome retraction following adhesion photocleavage may provide a model to understand how changes in tension can regulate function. While MT-FLIM has some current limitations in its temporal resolution, this method offers a unique solution to mapping forces at fluid interfaces and is a departure from past methods in its incorporation of a parameter that uniquely reports forces and is not subject to intramolecular fluorescence crosstalk. Photocleavable probes provide a new method to perturb individual adhesion structures with minimal disruption to the cell body and without changing the extracellular environment⁵¹

New Supplementary Figure:

Comment 2: The authors state in the introduction and discussion that they “demonstrate that podosome-podosome mechanical crosstalk is minimal”. They show that blebbistatin has no effect on integrin forces and conclude from this result that podosomes are not mechanically connected. However, supported lipid bilayers are laterally fluid and impair transmission of lateral forces. To clearly demonstrate that there is no crosstalk between podosomes in fibroblasts, experiments +/- blebbistatin should be performed, either on nano-partitioned rigid lipid bilayers (as described in Yu CH, Cell Reports 2013), or on rigid substrates.

Response: We realize that our manuscript may have been confusing, and we apologize for this inconvenience. While we do claim in our manuscript that podosomes in this system are not mechanically connected, this evidence comes from our experiments using photocleavable biotin in which we site-specifically cleave adhesions at a single or cluster of podosomes and tracked changes in tension and protrusion at podosome across the cell. Our blebbistatin data is used to make a slightly different claim – we observed no significant change in podosome radius and only

a slight change in percent open probes ($p=0.046$) upon drug treatment and thus concluded that integrin receptor force generation was independent of the myosin II pathway. Although $p < 0.05$, this p value was several magnitudes higher than that for the other drug treatments, and our analysis can be sensitive to the selected threshold, so we concluded this was not a meaningful biological difference. In the revised manuscript, these results are also supported with knockdown experiments. While the mechanical crosstalk experiments were distinct from these conclusions, the Reviewer raises an important point that a laterally fluid bilayer might impair force transmission across a network. We agree with this, but we counter that podosomes could not exhibit global alterations due to local changes in mechanics, because it is possible that there are other mechanisms at play such as long-range diffusion of adhesion proteins that could contribute. To address this, we have amended the manuscript as shown below.

We agree that the nanopatterned substrates would be interesting. However, these experiments are beyond the scope of this paper which focuses on developing the toolset to quantify podosome mechanics at the cell-membrane interface. Another issue is that the literature suggests that the nanopatterned grids inhibit podosome formation. Studying podosomes on glass would also be interesting for future work. We will not address it here since it requires a different type of cell model.

Revised Text (p. 11): *“We anticipated that optical release of DNA probes would terminate integrin tension and cause rapid refolding of the DNA hairpin and re-quenching of tension signal in the podosome ring (**Supplementary Fig. 16**). Upon 405 nm illumination of a $7 \mu\text{m}^2$ podosome-containing region, we tracked the changes in tension signal both at the site of photocleavage (proximal) and across the entire cell (distal). Although SLBs dissipate long-range forces, we still wondered if it would be possible to observe global changes that were communicated intracellularly, such as through altered diffusion of adhesion proteins³.”*

Comment 3: At the end of the introduction, the authors state that they “provide direct evidence that podosomes are mechanosensors with local pN sensitivity”. Podosome mechanosensitivity has been evaluated in other articles, but not in this manuscript. To conclude that, tension measurements should be performed with fibroblasts layered on substrates of controlled stiffness.

Response: While we appreciate the critique and careful assessment of our paper, we respectfully disagree with this comment. Indeed, other papers have tested the podosome mechanosensor model by using substrates of varying stiffness. In our paper, we investigate podosome mechanosensitivity at the molecular scale. The ability of podosomes to detect the molecular stiffness of the ECM is evaluated through the tension gauge tether experiments. These experiments show that the podosome senses piconewton changes in substrate mechanics. The tension gauge tether results demonstrate mechanosensation by altering the molecular stiffness of the ligand rather than the bulk stiffness of the substrate. This is now better explained in our text.

Revised Text (p. 12): *“The T_{tol} value can be tuned by changing probe geometry. In the “Unzipping Mode”, the cRGD ligand and biotin are on the same terminus of the duplex, and $T_{\text{tol}} = 12 \text{ pN}$. In the “Shearing Mode” the biotin anchor and the cRGD ligand are on opposite sides of the duplex, and $T_{\text{tol}} = 56 \text{ pN}$ (**Fig. 5b**). TGT experiments are unique in that they present chemically identical substrates that differ in their molecular stiffness. Thus, these experiments could also be used to test whether mechanosensory podosomes have piconewton sensitivity. We hypothesized that podosomes would be sensitive to these pN changes in ligand stiffness and that 12 pN TGTs would hinder podosome formation.”*

Comment 4: Results/ Podosome actin content correlates with RGD-probe depletion on SLBs: Fig1: “The depletion of RGD-DNA was caused by pushing forces generated by the protrusive podosome core”. This interpretation is not supported by the experiments. To ascertain that depletion is due to protrusion, the axial deformation of the SLB should be shown and quantified.

Response: We would like to thank the reviewer for their feedback, which highlighted the need for additional clarification and data in our paper. Our experiments clearly show that the RGD ligand is excluded out of the podosome core. This loss of RGD can be due to disruption of the lipid bilayer or it may be due to weaker forces that exclude the RGD ligand but maintain the membrane. Since the SLB is an incompressible material⁹ (further described in Response 4, Reviewer 2), it does not deform under these forces. Furthermore, no molecular compression force probes exist. Following, literature precedent from Sheetz and colleagues¹, as well as our own control experiments, we are confident that the depletion is due to pushing forces that exclude the RGD ligand. Our evidence comes from new membrane staining experiments and further quantification of our original data (**Supplementary Fig. 5**), as well as careful examination of our timelapse videos of podosome dynamics (**Supplementary Videos 1-3**). SLB compression can drive membrane collapse or phase transitions, and in all of our experiments we observe an intact membrane. Thus, and the podosome forces do not deform or disrupt the membrane.

Revised Text (p.5): *“The depletion of RGD-DNA was caused by pushing forces generated by the protrusive podosome core¹. Note that these forces exclude RGD-DNA but do not rupture the underlying SLB (**Supplementary Fig. 5**), which is largely incompressible^{8,9,11}.”*

New Supplementary Figure:

Comment 5: Results / MT-FLIM maps pN receptor tension and clustering on SLBs: “fluorescence lifetime is sensitive to energy transfer but independent of the dye concentration⁵⁰”. Can the authors rule out that there is no FRET between two probes as a result of a higher concentration of the probes around podosome cores? How does the calibration curve vary with probe concentration?

Response: Thank you for bringing this important caveat to our attention. While fluorescence lifetime of individual fluorophores is typically concentration independent below the threshold for homo-FRET between fluorophores, there is literature precedent for density dependent FRET probes (Ma, et. al., *Nat. Commun.*, 2017) and this is something that is important to explore for our work, too. To test this, we performed a new experiment where we incorporated varying levels of

biotinylated lipids (0.05 – 0.2 mol %) into our bilayer in order to modulate the probe density. We then performed epifluorescence and FLIM imaging on SLBs containing 0 – 20% open probes to test whether there were changes in fluorescence lifetime. We observed that the fluorescence lifetime did exhibit slight dependence on density (maximum change in lifetime = 0.13 nsec). The results are summarized in a new supplementary figure (**Supplementary Fig. 8**). This issue is now discussed in the results section. We would like to emphasize that these changes are small compared to the changes in fluorescence lifetime due to probe opening (~1.1 → 2.6 nsec) (Fig. 2).

Revised Results (p. 6): “We also characterized these SLBs using epifluorescence to determine probe quenching efficiency (**Fig. 2c**) and measured intensity and fluorescence lifetime as a function of probe density (**Supplementary Fig. 8**). We found a subtle decrease in probe lifetime with increasing probe density. However, this effect was minor compared to the effect of force-mediated hairpin opening.”

New Supplementary Figure:

Comment 6: Results / MT-FLIM maps pN receptor tension and clustering on SLBs: “Within 20 min of plating, cells formed dynamic integrin clusters which lacked detectable forces.”

These results are not shown. How were integrin clusters observed?

Response: We apologize for any data that may have been obscured in the supplementary packet. We include images and videos of clustering in Supplementary Figure 14. However, we have updated this figure and we now provide particle tracking of not only podosomes but also of receptor clusters. It was challenging to tell whether the data shows transient clusters or dynamic clusters and to assign clusters track identities. Therefore, to remove any human bias in our analysis, we segmented the images to mask clusters (estimated 95% accuracy) in MATLAB based on size and the lack of local depletion zone and then used Trackmate in FIJI to computationally assign lineage. For simplicity, tracking did not allow merging or splitting of trajectories, but these would be interesting topics of investigation for future works. These results are shown below.

Updated Supplementary Figure:

Comment 7: Results/ Actin protrusion and integrin tension are engaged in a mechanical feedback loop: “we engineered “releasable” MTFM probes by anchoring probes with a photocleavable (PC) biotin group”. As a control of the effect of the 405 nm laser, tension in podosomes should be measured in the absence of photocleavable biotin groups.

Response: We agree that this is a very important control, and we apologize if our figures and annotation was unclear. In efforts to resolve this, we have added a new schematic showing the different experimental and control probes used in these experiments. In addition, we have changed the labels in our related supplementary figures from “PC Biotin” and “Biotin” to “+ PC” and “- PC” to emphasize the difference between these groups. The most significantly changed figure is shown below, and these labels are carried out throughout the relevant supplementary figures. Note that we do observe some light-mediated disassembly at later time points, but these are minor at the quantified time points (-12 and 29 s)

Revised Text (p. 11): “Control experiments confirm that mechanical perturbations were largely responsible for the observed changes in signal, with phototoxicity contributing minimally at the 29 s time point. Note that at later time points, some cells do show some light-mediated podosome disassembly (Supplementary Figs. 21, 23).”

Revised Supplementary Figure:

Minor Comments:

Comment 1: The authors state that “pharmacological inhibitors show that forces are the direct consequence of actin-core polymerization”. However, the drugs used do not specifically target actin-core polymerization, but more generally the polymerization of actin in the whole cell. I suggest to reformulate this conclusion.

Response: Thank you for this comment. It is true that pharmacological inhibitors target the cytoskeleton and motor proteins across the entire cell and that the podosome is part of a complex actin network. Thus, it is possible that the drugs affect podosomes due to additional alterations in actin polymerization, particularly in the ring. We have softened our language from “actin core protrusion” to “actin protrusion” in the main text.

Comment 2: All along the manuscript, the authors use the term “contraction force” when mentioning the force applied at the podosome ring. “Contraction” implies that there is a (myosin II) contractility-based mechanism, which is in contradiction with the results of this manuscript since blebbistatin has no effect. I suggest to replace “contraction force” by “traction force”.

Response 2: This is an important point and one which we have also debated. In physics and engineering, traction force specifically refers to forces that are tangential to the substrate. Our work deals with vertical forces perpendicular to the substrate, and thus, we do not feel that traction force is the most appropriate choice of language. We had used a more colloquial definition of contractile forces, referring to any force generating contraction, but the reviewer explains, this term has a very technical definition referring to actomyosin contractility that is generated by myosin II. We now have selected to replace “contractile forces” with the term “tensile force.” We feel this is an appropriate selection for two reasons: 1) Tensile strength refers to the force to pull on a material. 2) There is an existing literature precedent to describe integrin pulling forces as tensile forces (Li and Springer, *PNAS*, 2016). Thus, our work now describes podosome protrusion forces and integrin tensile forces. The term contractile is still used in our text when it refers specifically to actomyosin contractility.

Comment 3: The term “organelles” should be replaced, e.g. by “structures”, as podosomes are not surrounded by lipid bilayers.

Response 3: We have updated the word from “organelles” to “structures.”

Comment 4: Fig3c: It would be interesting to evaluate how the angle varies with the distance to the actin core?

Response: This is a very interesting idea and one which we would like to explore in future works. Because podosome rings are on micron-scaled, they have been the target of many different super resolution imaging techniques^{37,38,52,53}. Unfortunately, molecular force microscopy is not currently compatible with super resolution imaging, which precludes our ability to measure with spatial resolution that is high enough to resolve the relationship between angle and radius.

Comment 5: Fig.6 is numbered Fig.8 in the text.

Response: We apologize for this inconsistency. We have resolved the labeling error.

Comment 6: “the nature of the mechanical linkage between the substrate and the cytoskeleton was unknown^{16,19,27,30,60}” In ref. 27, talin and vinculin, two proteins making mechanical links between integrins and actin have been identified.

Response: Thank you for your thorough evaluation of our work in the context of the literature. While we value your opinion, in this case we choose to respectfully disagree with this comment. Vinculin and talin are very important in podosome rings, but we counter that there is sufficient evidence to describe the mechanical linkage between the substrate and the cytoskeleton. While protein stretching is very plausible evidence of tension¹¹, it cannot directly report forces across the podosome ring complex. Furthermore, even in the case of tension across these adhesive proteins, without a receptor-level mechanical connection, one cannot rule out an alternate mechanism of force balance. This distinction is especially notable in a fluid supported lipid bilayer system in which it was previously thought that cells could not apply forces⁴⁶.

- 1 Yu, C. H. *et al.* Integrin-matrix clusters form podosome-like adhesions in the absence of traction forces. *Cell reports* **5**, 1456-1468, doi:10.1016/j.celrep.2013.10.040 (2013).
- 2 Labernadie, A., Thibault, C., Vieu, C., Maridonneau-Parini, I. & Charriere, G. M. Dynamics of podosome stiffness revealed by atomic force microscopy. *Proceedings of the National Academy of Sciences of the United States of America* **107**, 21016-21021, doi:10.1073/pnas.1007835107 (2010).
- 3 van den Dries, K. *et al.* Interplay between myosin IIA-mediated contractility and actin network integrity orchestrates podosome composition and oscillations. *Nature communications* **4**, 1412, doi:10.1038/ncomms2402 (2013).
- 4 Holoubek, A. *et al.* Monitoring of nucleophosmin oligomerization in live cells. *Methods and Applications in Fluorescence* **6**, 035016, doi:10.1088/2050-6120/aaccb9 (2018).
- 5 Anthony, N. R., Mehta, A. K., Lynn, D. G. & Berland, K. M. Mapping amyloid-beta(16-22) nucleation pathways using fluorescence lifetime imaging microscopy. *Soft matter* **10**, 4162-4172, doi:10.1039/c4sm00361f (2014).
- 6 Ostašov, P. *et al.* FLIM studies of 22- and 25-NBD-cholesterol in living HEK293 cells: Plasma membrane change induced by cholesterol depletion. *Chemistry and Physics of Lipids* **167-168**, 62-69, doi:<https://doi.org/10.1016/j.chemphyslip.2013.02.006> (2013).
- 7 Zwang, T. J., Fletcher, W. R., Lane, T. J. & Johal, M. S. Quantification of the Layer of Hydration of a Supported Lipid Bilayer. *Langmuir* **26**, 4598-4601, doi:10.1021/la100275v (2010).
- 8 Glazier, R. & Salaita, K. Supported lipid bilayer platforms to probe cell mechanobiology. *Biochimica et biophysica acta* **1859**, 1465-1482, doi:10.1016/j.bbamem.2017.05.005 (2017).
- 9 Evans, E. & Needham, D. Physical properties of surfactant bilayer membranes: thermal transitions, elasticity, rigidity, cohesion and colloidal interactions. *The Journal of Physical Chemistry* **91**, 4219-4228, doi:10.1021/j100300a003 (1987).
- 10 Almquist, B. D. & Melosh, N. A. Fusion of biomimetic stealth probes into lipid bilayer cores. *Proceedings of the National Academy of Sciences of the United States of America* **107**, 5815-5820, doi:10.1073/pnas.0909250107 (2010).
- 11 Boulbitch, A. *Deflection of a cell membrane under application of local force*. Vol. 57 (1998).
- 12 Takito, J., Inoue, S. & Nakamura, M. The Sealing Zone in Osteoclasts: A Self-Organized Structure on the Bone. *International journal of molecular sciences* **19**, doi:10.3390/ijms19040984 (2018).
- 13 Sage, P. T. *et al.* Antigen Recognition is Facilitated by Invadosome-Like Protrusions Formed by Memory/Effector T Cells. *Journal of Immunology (Baltimore, Md. : 1950)* **188**, 3686-3699, doi:10.4049/jimmunol.1102594 (2012).
- 14 Baranov, M. *et al.* Podosomes of dendritic cells facilitate antigen sampling. *Journal of cell science* **127**, 1052-1064, doi:10.1242/jcs.141226 (2014).
- 15 Hurst, I. R., Zuo, J., Jiang, J. & Holliday, L. S. Actin-related protein 2/3 complex is required for actin ring formation. *Journal of bone and mineral research : the official journal of the American Society for Bone and Mineral Research* **19**, 499-506, doi:10.1359/jbmr.0301238 (2004).
- 16 Poulter, N. S. *et al.* Platelet actin nodules are podosome-like structures dependent on Wiskott-Aldrich syndrome protein and ARP2/3 complex. *Nature Communications* **6**, 7254, doi:10.1038/ncomms8254 (2015).
- 17 Linder, S., Nelson, D., Weiss, M. & Aepfelbacher, M. Wiskott-Aldrich syndrome protein regulates podosomes in primary human macrophages. *Proceedings of the National Academy of Sciences of the United States of America* **96**, 9648-9653 (1999).
- 18 Buchbinder, D., Nugent, D. J. & Filipovich, A. H. Wiskott-Aldrich syndrome: diagnosis, current management, and emerging treatments. *The Application of Clinical Genetics* **7**, 55-66, doi:10.2147/TACG.S58444 (2014).

- 19 Calle, Y. *et al.* WASp deficiency in mice results in failure to form osteoclast sealing zones and defects in bone resorption. *Blood* **103**, 3552-3561, doi:10.1182/blood-2003-04-1259 (2004).
- 20 Raynaud-Messina, B. *et al.* Bone degradation machinery of osteoclasts: An HIV-1 target that contributes to bone loss. *Proceedings of the National Academy of Sciences of the United States of America* **115**, E2556-e2565, doi:10.1073/pnas.1713370115 (2018).
- 21 Verollet, C. *et al.* HIV-1 reprograms the migration of macrophages. *Blood* **125**, 1611-1622, doi:10.1182/blood-2014-08-596775 (2015).
- 22 Murphy, D. A. & Courtneidge, S. A. The 'ins' and 'outs' of podosomes and invadopodia: characteristics, formation and function. *Nature reviews. Molecular cell biology* **12**, 413-426, doi:10.1038/nrm3141 (2011).
- 23 Mandal, S., Johnson, K. R. & Wheelock, M. J. TGF-beta induces formation of F-actin cores and matrix degradation in human breast cancer cells via distinct signaling pathways. *Experimental cell research* **314**, 3478-3493, doi:10.1016/j.yexcr.2008.09.013 (2008).
- 24 Eddy, R. J., Weidmann, M. D., Sharma, V. P. & Condeelis, J. S. Tumor Cell Invadopodia: Invasive Protrusions that Orchestrate Metastasis. *Trends in cell biology* **27**, 595-607, doi:10.1016/j.tcb.2017.03.003 (2017).
- 25 Block, M. R. *et al.* Podosome-type adhesions and focal adhesions, so alike yet so different. *European journal of cell biology* **87**, 491-506, doi:10.1016/j.ejcb.2008.02.012 (2008).
- 26 Zhou, D. W. *et al.* Effects of substrate stiffness and actomyosin contractility on coupling between force transmission and vinculin–paxillin recruitment at single focal adhesions. *Molecular biology of the cell* **28**, 1901-1911, doi:10.1091/mbc.e17-02-0116 (2017).
- 27 Labernadie, A. *et al.* Protrusion force microscopy reveals oscillatory force generation and mechanosensing activity of human macrophage podosomes. *Nature communications* **5**, 5343, doi:10.1038/ncomms6343 (2014).
- 28 Kronenberg, N. M. *et al.* Long-term imaging of cellular forces with high precision by elastic resonator interference stress microscopy. *Nature cell biology* **19**, 864-872, doi:10.1038/ncb3561 (2017).
- 29 Proag, A., Bouissou, A., Vieu, C., Maridonneau-Parini, I. & Poincloux, R. Evaluation of the force and spatial dynamics of macrophage podosomes by multi-particle tracking. *Methods (San Diego, Calif.)* **94**, 75-84, doi:10.1016/j.ymeth.2015.09.002 (2016).
- 30 van den Dries, K., Bolomini-Vittori, M. & Cambi, A. Spatiotemporal organization and mechanosensory function of podosomes. *Cell Adh Migr* **8**, 268-272 (2014).
- 31 Gupta, M., Doss, B., Lim, C. T., Voituriez, R. & Ladoux, B. Single cell rigidity sensing: A complex relationship between focal adhesion dynamics and large-scale actin cytoskeleton remodeling. *Cell Adh Migr* **10**, 554-567, doi:10.1080/19336918.2016.1173800 (2016).
- 32 Zhang, Y. *et al.* Platelet integrins exhibit anisotropic mechanosensing and harness piconewton forces to mediate platelet aggregation. *Proceedings of the National Academy of Sciences of the United States of America* **115**, 325-330, doi:10.1073/pnas.1710828115 (2018).
- 33 Galior, K., Liu, Y., Yehl, K., Vivek, S. & Salaita, K. Titin-based Nanoparticle Tension Sensors Map High-Magnitude Integrin Forces within Focal Adhesions. *Nano letters* **16**, 341-348, doi:10.1021/acs.nanolett.5b03888 (2016).
- 34 Zhang, Y., Ge, C., Zhu, C. & Salaita, K. DNA-based digital tension probes reveal integrin forces during early cell adhesion. *Nature communications* **5**, 5167, doi:10.1038/ncomms6167 (2014).
- 35 Liu, Y. *et al.* Nanoparticle tension probes patterned at the nanoscale: impact of integrin clustering on force transmission. *Nano Lett* **14**, 5539-5546, doi:10.1021/nl501912g (2014).
- 36 Plotnikov, S. V., Sabass, B., Schwarz, U. S. & Waterman, C. M. High-Resolution Traction Force Microscopy. *Methods in cell biology* **123**, 367-394, doi:10.1016/B978-0-12-420138-5.00020-3 (2014).

- 37 van den Dries, K. *et al.* Dual-color superresolution microscopy reveals nanoscale organization of mechanosensory podosomes. *Molecular biology of the cell* **24**, 2112-2123, doi:10.1091/mbc.E12-12-0856 (2013).
- 38 Bouissou, A. *et al.* Podosome Force Generation Machinery: A Local Balance between Protrusion at the Core and Traction at the Ring. *ACS nano* **11**, 4028-4040, doi:10.1021/acsnano.7b00622 (2017).
- 39 Liu, Z. *et al.* Mechanical tugging force regulates the size of cell–cell junctions. *Proceedings of the National Academy of Sciences* **107**, 9944 (2010).
- 40 Kedziora, K. M., Isogai, T., Jalink, K. & Innocenti, M. Invadosomes - shaping actin networks to follow mechanical cues. *Frontiers in bioscience (Landmark edition)* **21**, 1092-1117 (2016).
- 41 Linder, S. & Wiesner, C. Feel the force: Podosomes in mechanosensing. *Experimental cell research* **343**, 67-72, doi:10.1016/j.yexcr.2015.11.026 (2016).
- 42 Luxenburg, C., Winograd-Katz, S., Addadi, L. & Geiger, B. Involvement of actin polymerization in podosome dynamics. *Journal of cell science* **125**, 1666-1672, doi:10.1242/jcs.075903 (2012).
- 43 Revach, O.-Y. *et al.* Mechanical interplay between invadopodia and the nucleus in cultured cancer cells. *Scientific Reports* **5**, 9466, doi:10.1038/srep09466
<https://www.nature.com/articles/srep09466#supplementary-information> (2015).
- 44 Hu, S. *et al.* Podosome rings generate forces that drive saltatory osteoclast migration. *Molecular biology of the cell* **22**, 3120-3126, doi:10.1091/mbc.E11-01-0086 (2011).
- 45 Rafiq, N. B. M. *et al.* Podosome assembly is controlled by the GTPase ARF1 and its nucleotide exchange factor ARNO. *The Journal of Cell Biology* **216**, 181-197, doi:10.1083/jcb.201605104 (2017).
- 46 Bennett, M. *et al.* Molecular clutch drives cell response to surface viscosity. *Proceedings of the National Academy of Sciences of the United States of America* **115**, 1192-1197, doi:10.1073/pnas.1710653115 (2018).
- 47 Ma, V. P.-Y. *et al.* Ratiometric Tension Probes for Mapping Receptor Forces and Clustering at Intermembrane Junctions. *Nano Letters* **16**, 4552-4559, doi:10.1021/acs.nanolett.6b01817 (2016).
- 48 Nowosad, C. R., Spillane, K. M. & Tolar, P. Germinal center B cells recognize antigen through a specialized immune synapse architecture. *Nature immunology* **17**, 870-877, doi:10.1038/ni.3458 (2016).
- 49 Yu, C.-h. *et al.* Integrin-beta3 clusters recruit clathrin-mediated endocytic machinery in the absence of traction force. *Nature communications* **6**, 8672-8672, doi:10.1038/ncomms9672 (2015).
- 50 Pourfarhangi, K. E., Bergman, A. & Gligorijevic, B. ECM Cross-Linking Regulates Invadopodia Dynamics. *Biophysical journal* **114**, 1455-1466, doi:10.1016/j.bpj.2018.01.027 (2018).
- 51 Mohd Rafiq, N. *et al.* Forces and constraints controlling podosome assembly and disassembly. *bioRxiv*, 495176, doi:10.1101/495176 (2018).
- 52 Cox, S. *et al.* Bayesian localization microscopy reveals nanoscale podosome dynamics. *Nature methods* **9**, 195-200, doi:10.1038/nmeth.1812 (2011).
- 53 Walde, M., Monypenny, J., Heintzmann, R., Jones, G. E. & Cox, S. Vinculin binding angle in podosomes revealed by high resolution microscopy. *PLoS one* **9**, e88251-e88251, doi:10.1371/journal.pone.0088251 (2014).

Reviewers' Comments:

Reviewer #1:

Remarks to the Author:

The authors have carefully addressed all my concerns. I thus now recommend publication.

Reviewer #2:

Remarks to the Author:

The authors have done an excellent job in addressing the remaining concerns. The data are convincing and well controlled, and the manuscript is written very well. I therefore recommend the publication of this study.

Reviewer #3:

Remarks to the Author:

Review of revised manuscript by Glazier et al. "DNA mechanotechnology reveals that integrin receptors apply pN forces in podosomes on fluid substrates".

The authors have adequately addressed all the questions pointed out by the previous review, improving the manuscript and clarifying all the puzzling points of the original version. Hence, we now recommend the publication of the revised manuscript in Nature Communications.